# AMAGO-2: Breaking the Multi-Task Barrier in Meta-Reinforcement Learning with Transformers

**Jake Grigsby**  **Justin Sasek**[†]  **Samyak Parajuli**[†]  **Daniel Adebi**[†]  **Amy Zhang**  **Yuke Zhu**
The University of Texas at Austin
† Equal contribution
{grigsby,yukez}@cs.utexas.edu

## Abstract

Language models trained on diverse datasets unlock generalization by in-context learning. Reinforcement Learning (RL) policies can achieve a similar effect by meta-learning within the memory of a sequence model. However, meta-RL research primarily focuses on adapting to minor variations of a single task. It is difficult to scale towards more general behavior without confronting challenges in multi-task optimization, and few solutions are compatible with meta-RL's goal of learning from large training sets of unlabeled tasks. To address this challenge, we revisit the idea that multi-task RL is bottlenecked by imbalanced training losses created by uneven return scales across different tasks. We build upon recent advancements in Transformer-based (in-context) meta-RL and evaluate a simple yet scalable solution where both an agent's actor and critic objectives are converted to classification terms that decouple optimization from the current scale of returns. Large-scale comparisons in Meta-World ML45, Multi-Game Procgen, Multi-Task POPGym, Multi-Game Atari, and BabyAI find that this design unlocks significant progress in online multi-task adaptation and memory problems without explicit task labels.

## 1 Introduction

Billion-parameter generative models trained to imitate human language and behaviors from web datasets have unlocked unprecedented generality in machine learning [1, 2]. While evaluations of these large language models (LLMs) are organized into discrete benchmarks [3] with specific themes like competitive mathematics [4] or reading comprehension [5], their knowledge transfers to countless problems. LLMs' flexibility is largely due to an "in-context learning" effect that emerges when input sequences grow long enough to resemble a dataset of examples for a new task [6, 7]. However, predicting the most likely continuation of a sequence is misaligned with optimal control and limits us to tasks where high-quality demonstrations are available.

Online Reinforcement Learning (RL) lets models continuously self-improve and directly optimizes performance. The RL equivalent of in-context learning is a subset of meta-RL techniques that use sequence models for memory and adaptation [8, 9, 10]. Meta-RL can be viewed as a *task inference* problem in which an agent explores its new surroundings to discover information that it can use to improve decision-making [11, 12, 13]. In theory, meta-RL can adapt to a wide range of control problems. However, current applications are quite narrow — usually focusing on minor variations of what could be considered a single task. For example, imagine a platformer video game with procedurally generated levels. The game has a consistent theme and controls, but each level introduces a slightly different reward function, physics, and layout. A meta-learning policy trained on this game would learn to identify the relevant changes in a new level in order to maximize returns. By most formal definitions, any two levels from this game would be considered two different "tasks," but it seems unlikely that anyone outside the field would refer to the resulting policy as a "multi-task" agent.

38th Conference on Neural Information Processing Systems (NeurIPS 2024).

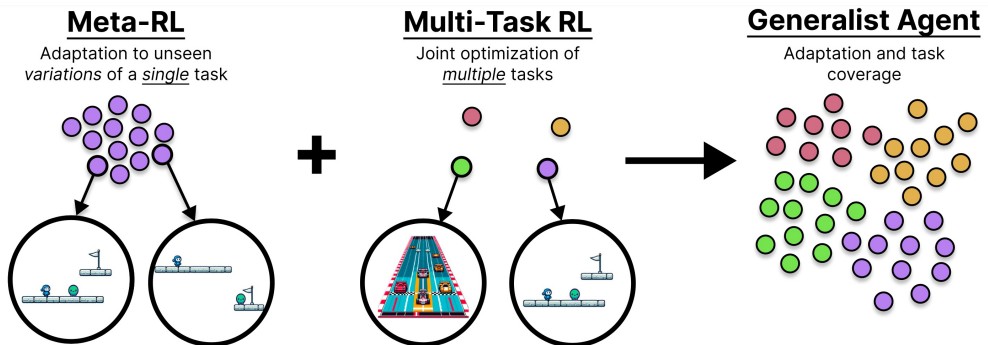

Figure 1: **Task Spaces in RL Generalization.** Meta-RL agentso adapt to dense variations of a core task. Multi-Task RL overcomes optimization challenges of learning from isolated tasks. Scalable ideas from both areas allow us to extend adaptive agents towards increasingly general behavior.

A truly general agent would also be able to play entirely different games. We will use the term "task" to refer to this kind of multi-game generality while calling different levels of the same game "task *variants*" [14, 15, 16]. Each task (game) creates a distribution of variants (levels) we can sample. The distinction between two "tasks" and two "variants" of the same task is arbitrary in theory[1]. However, it is so relevant in practice that having separate terms will be important for clarity.

Ideally, we would train RL agents on many different tasks and many variants of those tasks. Recent works have scaled to heterogeneous mixtures of popular RL benchmarks [19, 20, 21], but rely on offline datasets collected by expert single-task agents. Scaling up online learning is challenging because multi-task distributions can create disconnected datasets; as we transition from a single task to small sets of several tasks, we reach a barrier where the gap between tasks becomes so wide that practical optimization challenges hinder learning more than any knowledge transfer between them is helping [15, 22, 23]. This conflict is particularly limiting in meta-RL because multi-task behavior should otherwise be a simple case of task inference. Meta-RL agents learn to infer the identity of task variants based on limited information — which becomes more challenging as those variants become more similar. Adding an entirely new task with clearly distinct visuals or controls does little to the difficulty of this problem, and yet it can dramatically reduce performance because it introduces the (mostly unrelated) challenge of optimizing a second reward function. Fortunately, this challenge is not unique to meta-RL and is a focus of broader research in multi-task RL (MTRL) and supervised learning [24]. MTRL methods can improve multi-objective training by creating task-specific network parameters or editing the gradients of each task to resolve conflicting updates (Section 2). Unfortunately, these solutions rely on our ability to manually label tasks, scale with the size of our training set, and can abandon meta-RL's goal of adapting to new tasks at test-time. While MTRL might be practical on current benchmarks, it would be more scalable to preserve meta-RL's task inference and improve multi-task optimization without introducing new assumptions (Figure 1).

This paper focuses on training sequence model policies that generalize across hybrid task distributions containing elements of meta-RL, MTRL, and long-term memory. We revisit the idea that MTRL is primarily bottlenecked by imbalanced training losses created by tasks with returns on significantly different scales [25]. However, we approach this problem from a meta-RL perspective where *task inference needs to be learned; we do not allow knowledge of task labels to influence training in any way*. We experiment with Transformer [26] policies trained by mixtures of actor and critic objectives that are either directly or indirectly dependent on the scale of returns in any given task. Comparisons on Meta-World ML45 [17], Multi-Task POPGym [27], Multi-Game Procgen [28], Multi-Game Atari [29], and Multi-Task BabyAI [30] evaluate the importance of scale-resistant updates. We find that converting value regression to classification [31] and policy improvement to filtered imitation learning [32] leads to significant improvements in these hybrid settings. Our results suggest that scale invariance should be a priority when designing multi-task agents with adaptive memory. Our empirical success with classification updates further supports recent observations that sequence-based RL can improve by borrowing technical details from supervised sequence modeling [33, 34, 35] while retaining the online self-improvement process that makes RL useful [36, 37].

---

[1]The distinction between tasks/variants is also called non-parametric/parametric variation [17, 18]. Variants are often generated by randomizing a known set of parameters in a single (simulated) environment.

## 2 Background

**Memory-Based Meta-RL.** The objective of a meta-RL agent is to adapt its policy to maximize returns over a horizon of one or more attempts in a new task. We focus on a subset of "black-box" approaches that reduce meta-learning to the problem of training sequence model policies with end-to-end RL [8]. We view adaptation as a form of partial observability [12], where instead of inferring the current state $s_t$ from a sequence of observations $(o_0, \ldots, o_t)$, we try to infer the state *and* the identity of the task from trajectories ($\tau$) of observations, actions ($a$), and rewards ($r$): $\tau_{0:t} = (o_0, a_0, r_0, o_1, \ldots, a_{t-1}, r_{t-1}, o_t)$ [13]. A task's identity is an unobservable variable describing aspects of its dynamics that vary across the training distribution [38]. Because improved estimates of the current task would increase returns, meta-reasoning can arise implicitly in the latent space of a sequence model that is trained by standard RL updates.

This sequence-based meta-RL framework, which began with RL$^2$ [9, 10, 39], has two key advantages. First, training policies on trajectory sequences turns standard partial observability and zero-shot generalization [40] into special cases. In practice, this creates flexible agents that blur formal boundaries between memory, generalization, and meta-learning [41]. Second, end-to-end RL enables self-improvement without demonstrations. Many meta-RL algorithms improve performance in applications where task identification could benefit from additional assumptions [11, 42, 43, 44]. Another line of work recovers the flexibility of "in-context" sequence learning when a dataset of demonstrations makes self-improvement unnecessary [45, 46, 47, 48]. Most of the algorithmic variety in meta-RL comes from these two areas. In contrast, the RL$^2$ approach has primarily become an engineering problem. Important developments include increasing the adaptation horizon by enabling long-context sequence models in an RL setting [49, 50, 51], and a shift away from on-policy updates in favor of off-policy actor-critics [52, 53, 54]. Combinations of these details allow Transformers with long-term recall to self-improve on large datasets of recycled data [55, 56, 57].

**Multi-Task Optimization.** Multi-task learning addresses a trade-off where conflicting objectives decrease performance relative to single-task learning despite an increase in training data. One approach uses specialized optimizers that edit gradients to balance tasks' local optima [22, 58, 59, 60, 61]. However, it can be as effective (and less expensive) to rescale each task's contribution to the overall loss [23, 25, 62, 63, 64]. These techniques require the ability to identify the current task, which is commonly available in multi-task settings: we can view MTRL as a meta-RL problem where the ground-truth identity of each environment has been one-hot labeled in advance. MTRL often depends on task labels in more subtle ways, including networks with task-specific output heads [65] and datasets that balance sampling [66] or normalize rewards [67] across tasks.

**Problem Setting.** We focus on hybrid generalization problems that require elements of meta-learning, multi-task learning, and long-term memory. Meta-RL often evaluates performance over $k > 1$ attempts or episodes. A "$k$-episode" objective maximizes the *total* return over all $k$ attempts and creates an exploration/exploitation trade-off at test-time [13]. A "$k$-shot" objective allows for an exploration window where only the final episode counts towards the agent's score [42, 68]. We measure generality by dividing a set of task variants ($\mathcal{V}$) into train and test sets. Meta-RL benchmarks are (informally) characterized by their relative density — meaning $|\mathcal{V}|$ is large and composed of subtle changes to a common objective. Multi-task problems are created by training a single policy to maximize $N$ qualitatively distinct objectives, such as learning to play $N$ Atari games [29]. More general settings might be described as containing $N$ adaptation problems with their own set of variants $\mathcal{V}_0, \ldots, \mathcal{V}_N$.

**Scaling Meta-RL Beyond the Multi-Task Barrier.** Early meta-RL experiments focused on adapting to minor variations of multi-armed bandits [9], gridworlds [68], and locomotion benchmarks [43, 69, 70]. Despite significant algorithmic progress, these domains are still representative of recent research [44, 45, 54, 71]. Meta-RL requires generating many variations of a core problem while keeping those changes partially observed; it is difficult to design a cohesive benchmark that supports the dense level of variety it takes to induce meta-learning [55, 72]. We can create toy meta-RL problems, but there is a wide gap in complexity before we reach naturally occurring domains where long-term adaptation is necessary. For example, one application of meta-RL would be to learn to master unfamiliar video games over several minutes or hours. Building such an agent would likely involve scaling to a large training set of games. It is difficult to make gradual progress towards

this goal because it begins as a memory problem, becomes a multi-task problem, and then only requires meta-learning at an extreme scale. Individual games often require long-term memory, but as we add games, there will inevitably be isolated groups with unique objectives that introduce challenges from MTRL. Eventually, we reach a level of training diversity where adaptation becomes critical and meta-learning emerges. The central issue considered by this paper is that multi-task optimization limits train-time performance at task counts far smaller than would be necessary for test-time generalization to new tasks. A promising direction is to take a method that is already capable of both memory and meta-learning and then find a way to break the barrier where progress is limited by multi-task optimization. However, if we want to adapt to many tasks, we should avoid solutions that scale with the number of training tasks or depend on our ability to identify them manually.

This "multi-task barrier" impacts both long-term applications and current research benchmarks. For a more practical example, consider Meta-World [17]. Meta-World is a suite of 50 robotic manipulation tasks. Each task creates a meta-RL problem by adjusting an unobservable goal so that agents must interpret incoming reward signals to adapt to the current objective. Meta-World ML45 extends generalization by merging meta-RL tasks into a multi-task training set $\cup_{i=0}^{N=45}\mathcal{V}_i$. Identifying the current task during training should be a special case of meta-RL because the changes between tasks are less ambiguous than the changes in goal location between variants of the same task. For example, an agent might distinguish a pick-and-place task from a door opening task after a single observation of the objects on the table. ML45 reserves five tasks to benchmark meta-RL's long-term goal of adapting to entirely new tasks at test-time. However, learning task adaptation from just $N = 45$ training tasks is not realistic without additional assumptions [17, 73]. ML45 is clearly bottlenecked by MTRL: a similar setup *with the meta-RL component removed* is a challenging MTRL benchmark (MT50) by itself [59, 74]. MTRL methods use task labels to balance loss functions, separate network architectures, or edit task gradients. However, these techniques scale with $N$ and require a formal identification of each "task" that meta-RL otherwise should not need. Our goal is to overcome the same challenge while preserving the meta-RL perspective that task differences are arbitrary so that we can scale to unstructured domains where $N$ is large [19] or infinite. Therefore, even though our experiments will have a clearly defined $N$, we will enforce a strict constraint where the agent can have zero knowledge of which task is active or how many tasks there are $(N, |\mathcal{V}_n|)$.

## 3   Multi-Task Adaptation Without Task Labels

We study multi-task adaptation without task labels by building on the flexible memory-based framework in which sequence models optimize standard RL objectives across trajectory inputs (Section 2). Agents observe trajectory slices from a replay buffer with a context length of up to $l$ timesteps ($\tau_{t-l:t}$). The information revealed between timesteps ($o_i, a_{i-1}, r_{i-1}, d_{i-1}$) is merged into a single embedding $o_i^+$. The sequence ($o_{t-l}^+, \ldots, o_t^+$) is then passed through a sequence-to-sequence model to produce ($h_{t-l}, \ldots, h_t$) which serves as a shared representation for output heads representing the critic(s) $Q$ and stochastic policy $\pi$ (Figure 2). We focus on the combination of Transformers and off-policy RL updates, which will let us evaluate large-scale policies trained over millions of timesteps. Let $y_t = r_t + \gamma\bar{Q}(h_{t+1}, a' \sim \bar{\pi}(h_{t+1}))$ be the one-step temporal difference target, where $\bar{Q}$ and $\bar{\pi}$ refer to frozen target networks [75]. We can minimize standard off-policy actor-critic loss terms [75, 76] in parallel over the context length:

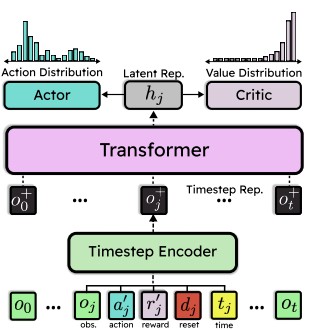

Figure 2: **Transformer-based Actor-Critic Architecture.**

$$\mathcal{L}_{\text{Critic}}(t) = (Q(h_t, a_t) - y_t)^2 \quad (1) \qquad \mathcal{L}_{\text{Actor}}(t) = -Q\big(h_t, a \sim \pi(h_t)\big) \quad (2)$$

Following details in AMAGO [57], we compute Equations (1) and (2) with an ensemble of critics [77] and over several different values of the discount factor $\gamma$ in parallel [78]. By preventing the parameters of $Q$ from minimizing $\mathcal{L}_{\text{Actor}}$, we can update the full architecture with one step of the joint loss: $\sum_{i=t-l}^{t-1}\mathcal{L}_{\text{Actor}}(i) + \lambda\mathcal{L}_{\text{Critic}}(i)$, where $\lambda$ is a hyperparameter that balances the objectives for the shared sequence model [56, 57].

Multi-task optimization challenges are not unique to RL [22, 62], but RL typically adds a bottleneck that we can remove: both Equations (1) and (2) directly depend on the scale of returns $Q$ in each of our $N$ tasks. When computed as a simple expectation over tasks, the gradients of $\mathcal{L}_{\text{Critic}}$ and $\mathcal{L}_{\text{Actor}}$

can favor small *relative* improvements in tasks with high *absolute* returns (Figure 3 left). We can almost never assume tasks have similar return scales. For example, even when we design reward functions to have similar initial and optimal returns, differences in task difficulty lead to uneven learning progress and imbalanced $Q$-values. This issue was proposed and demonstrated by methods such as Multi-Task PopArt [25]. PopArt's solution is to compute MTRL loss terms on a relative basis by creating a new critic output for each of the $N$ tasks. Each critic adaptively normalizes $y$ for its task and automatically rescales its output to help compensate for shifting targets. Simpler versions of this approach (e.g., without network rescaling) are universal in MTRL. However, we cannot use this technique without ground-truth knowledge of task labels. Although we cannot adaptively balance each task's loss, we can reformulate Eqs. (1) and (2) to be less sensitive to the scale of $Q$.

**Scale-Resistant Critics.** Distributional RL [79] is motivated by representation learning and risk-aware policies, but methods like C51 [80] have a useful side-effect of turning (1) into a classification problem that scales with the number of output bins instead of $Q$. For this reason, C51-style critic updates have become a key implementation detail in large-scale (offline) MTRL agents [65, 35]. We can keep the classification side-effect without modeling return distributions by converting a scalar $Q$ to discrete label(s) $\in \{0, \ldots, B\}$. Two-hot labels create a one-to-one mapping between every scalar in a fixed range and a discrete distribution where all non-zero probability is placed in two adjacent bins [81, 82]. If $Q_B$ denotes a modified critic network that outputs probabilities over $B$ bins, the TD-error can be minimized by multi-label classification:

$$\mathcal{L}_{\text{Critic-Ind}}(t) = -\text{twohot}_B(y_t)^\mathsf{T} \log Q_B(h_t, a_t) \tag{3}$$

We can reduce tuning of label spacing and limits with an invertible transform that maps a wide range of returns to a smaller number of labels [83]; we use `symlog` as demonstrated by DreamerV3 [31] and TD-MPC2 [20]. We also experimented with using a global (task-agnostic) $Q$-normalization for this purpose but found that a fixed mapping is important because stabilizing learning against shifting labels required extensive hyperparameter tuning. Figure 3 visualizes why we would expect classification to be an improvement in a multi-task setting: $\mathcal{L}_{\text{Critic-Ind}}$ maps the same relative error in value prediction to a similar loss value over a

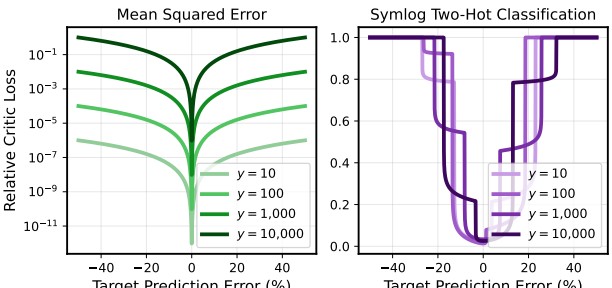

Figure 3: **Scale-Resistant Value Regression.** We plot the value of the standard critic loss (Eq. 1) as a function of the *relative* prediction error of the TD target ($y$) across four orders of magnitude (left). Y-axes are self-normalized according to the largest displayed value. Two-Hot classification (Eq. 4) maps the same relative error to similar loss values across the different absolute return scales of each task (right).

wide range return scales. In contrast, $\mathcal{L}_{\text{Critic}}$ can bias optimization towards tasks that happen to have large regression targets. This effect can be dependent on proper tuning of the bin count $B$, upper/lower return bounds, and use of the `symlog` transform (Appendix A).

A concurrent work [34] evaluates the two-hot critic loss' ability to unlock scaling laws in RL when used as a direct replacement for $\mathcal{L}_{\text{Critic}}$ (1) (or its algorithm-specific equivalent). They find the classification loss improves performance in a range of domains and attribute this to advantages in representation learning and robustness to noisy value targets ($y$). Our experiments focus on the PopArt hypothesis that scale-invariance improves optimization in an online multi-task setting. We will be exploring the impact of classification losses in an actor-critic agent with adaptive memory, where we also need to consider the scale of the policy's objective (2).

**Scale-Resistant Actors.** Many policy updates resemble a weighted regression objective where the importance of each action label is scaled by a function $f(\tau, a)$ that increases with its value or advantage, such as $f(\tau, a) \propto \exp(A(\tau, a))^2$. Methods differ in where the candidate actions $a$ are sampled from, how values are estimated, and how $f(\tau, a)$ clips or rescales the update [82, 84, 85,

---

[2]Where $A(\tau, a) = Q(\tau, a) - Q(\tau, a' \sim \pi(\tau))$ is the improvement of $a$ over the value of the current policy as estimated by actor and critic networks that take trajectory sequences as input.

86, 87]. In the off-policy setting where actions are sampled from a dataset of previous experience, slight differences in advantage estimates and weight functions $f$ create a dense family of similar advantage-weighted regression (AWR) methods [86, 87, 88, 89, 90]. AWR is intuitive and simple to implement but directly depends on the scale of returns unless we intentionally avoid this with a binary filter $f(\tau, a) = \mathbb{1}\{A(\tau, a) > 0\}$ [32, 91]. We follow details from CRR-Binary-Mean [32] for a candidate actor loss that is independent of $Q$:

$$\mathcal{L}_{\text{Actor-Ind}}(t) = -\mathbb{1}\{Q(h_t, a_t) - \underset{a' \sim \pi(h_t)}{\mathbb{E}}[Q(h_t, a')] > 0\} \log \pi(a_t \mid h_t) \tag{4}$$

Intuitively, CRR performs imitation learning (IL) on action labels that we estimate will improve the current policy. Advantage estimates within $\mathcal{L}_{\text{Actor-Ind}}$ rely entirely on one-step dynamic programming (1) so that they never become outdated [32, 92]. Because they do not allow the policy to extrapolate to out-of-distribution actions, weighted IL updates like CRR are best suited to *offline* RL [93]. In offline RL, the ability to stabilize optimization on static datasets is well worth losing the optimistic exploration of $\mathcal{L}_{\text{Actor}}$ (2). Choosing to use $\mathcal{L}_{\text{Actor-Ind}}$ online implies that the ability to address issues in multi-task optimization is a similarly worthwhile trade-off. There are other RL engineering factors to consider in the specific case of training long-context Transformers. For example, sampling batches of trajectory sequences can create replay ratios high enough to effectively be offline RL [94]. However, the simplest justification for training Transformers on $\mathcal{L}_{\text{Actor-Ind}}$ may come from its similarities to widely used methods for supervised learning in RL.

**Transformers and RL via Supervised Learning.**   Decision Transformer (DT) is a popular approach to training sequence policies that reformulates RL as IL conditioned on a target return [36, 95, 37]. DT's advantage is that it reduces to IL on expert datasets, making it a safe improvement over a technique that is already effective for many problems. In a field where engineering is critical, and results can be hard to reproduce, there is valuable simplicity in inheriting technical details from supervised sequence modeling. This simplicity has likely contributed to the method's success despite several disadvantages. For example, DT policies are conditioned on the optimal return in order to imitate the best continuation of the current trajectory, but $\mathcal{L}_{\text{Actor-Ind}}$ does this without needing to define the optimal return in advance. DT determines the value of an action based on the fixed return of its trajectory, while $\mathcal{L}_{\text{Actor-Ind}}$ continuously updates its estimate with the value of the current network. By extension, DT cannot handle stochastic reward functions [96]. Like DT, $\mathcal{L}_{\text{Actor-Ind}}$ becomes IL on expert datasets, and Eqs. (3) and (4) resemble supervised learning with two classification heads.

## 4   Experiments

$\mathcal{L}_{\text{Actor}}$, $\mathcal{L}_{\text{Critic}}$, $\mathcal{L}_{\text{Actor-Ind}}$, and $\mathcal{L}_{\text{Critic-Ind}}$ create four interchangeable combinations of learning updates where one or both of the actor and critic loss can be dependent or *indirectly* dependent on the scale of $Q$. We will compare all four update variants, although we are mainly interested in evaluating the two extremes: "Dep. Actor, Dep. Critic" ($\mathcal{L}_{\text{Actor}}$, $\mathcal{L}_{\text{Critic}}$) and "Ind. Actor, Ind. Critic" ($\mathcal{L}_{\text{Actor-Ind}}$, $\mathcal{L}_{\text{Critic-Ind}}$). Our experimental setup allows for a direct ablation where all other details can be held fixed. Appendix A includes additional implementation details. Our experiments focus on three main questions: **1)** Do scale-resistant actor-critic objectives offer empirical benefits in challenging generalization problems? **2)** If so, can these gains be attributed to multi-task return scaling? and **3)** What new applications of memory-based policies might be unlocked by improvements in multi-task optimization without task labels?

**Meta-World ML45.**   ML45 is a challenging suite of 45 robotic manipulation meta-RL tasks. We argue that much of ML45's difficulty can be attributed to variance in return scales: ML45's training tasks are designed to have similar optimal returns, but they have different levels of difficulty, which causes the scale of $Q$-values to diverge early in training. The PopArt method of normalizing $Q$ per task may help balance optimization. Methods built on the original Meta-World results normalize the returns of each task separately [67], which is a simpler approach to the same idea. We force all task identification to be learned within the context of our memory policy so that one-hot labels and the total task size (45) do not influence training. The training environment samples a new task between meta-rollouts and the task identity is not included in the observation. Every task shares a single critic head, and all experience is added to a shared replay buffer sampled uniformly at random.

Figure 4 compares the four combinations of learning updates described in Section 3. Actor and critic loss terms that do not scale directly with the $Q$-values of each task improve skill coverage,

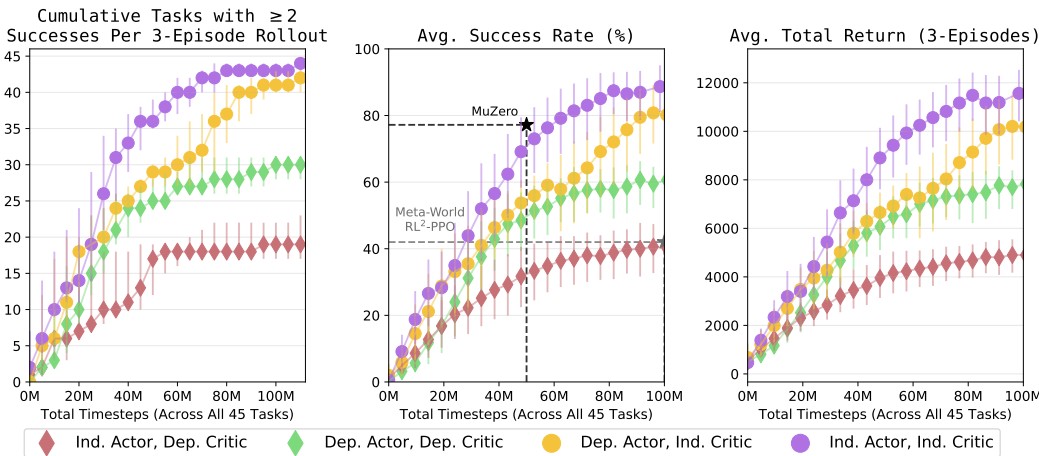

Figure 4: **Meta-World ML45 Train Task Results. (Left)** Coverage of the 45 manipulation skills measured by an adaptation horizon success rate $\geq 2/3$. **(Center)** Average success rate over tasks, variants, and 3-episode rollouts. Reference scores for MuZero and RL$^2$-PPO are gathered from results in [17, 73]. **(Right)** Total return over a 3-episode adaptation horizon, averaged across tasks and variants. All error bars indicate the maximum and minimum metric across three random trials.

overall success rate, and cumulative return across 3-episode adaptation horizons. Replacing $Q$ regression with two-hot classification appears to be the most important change, although substituting the weighted IL policy update improves performance and sample efficiency. Full learning curves for all 45 tasks are plotted in Appendix C and highlight the uneven learning progress (and therefore $Q$-value scale) of tasks throughout training. With a sample budget of 100M timesteps, our off-policy Transformers trained by max likelihood losses ($\mathcal{L}_{\text{Actor-Ind}}, \mathcal{L}_{\text{Critic-Ind}}$) more than double the success rate of Meta-World's original RL$^2$ result over the same sample limit while maintaining a $1.4\times$ improvement on its performance at 400M timesteps. Our simple one-step $Q$-learning matches more complex value updates like MuZero [81] at its reported 50M timestep budget [73]. We use a general Transformer architecture that would be equally applicable to any sequence modeling problem, but our results are also comparable to recent work such as HTrMRL [97] (success rate $\approx 85\%$) that use architectures specialized for multi-episodic RL.

**Multi-Task POPGym.** Meta-World is designed as a meta-learning benchmark, but its adaptation does not require long-term memory. POPGym [27] is a suite of zero-shot POMDP problems. Many POPGym tasks are explicit tests of long-term recall, but others involve state and variant estimation from previous actions and rewards. We use a simple observation and action space padding approach to create a multi-task version of POPGym: we combine all 27 tasks where both the action space and observation space have dimension $< 30$. It is not clear whether zero-shot performance in this setting can be compared against single-task POPGym results, as a strong agent may need to lose valuable time identifying which of the 27 memory games it is currently playing. Therefore, we evaluate in a one-shot setting where the first trial is a free exploration window used to determine which task is active so that the second trial creates a fair comparison to single-task results [68]. Importantly, we avoid "spoiling" any memory challenges for the second trial by randomly resetting to a new variant of the same task.

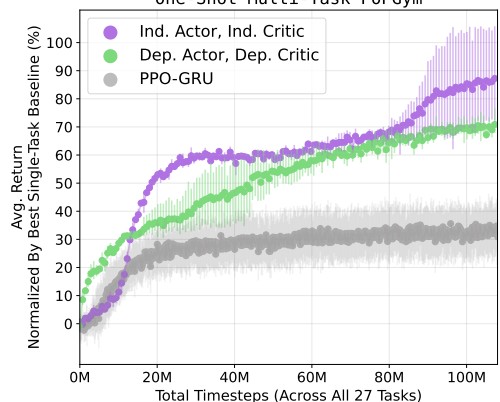

Figure 5: **Multi-Task POPGym Results.** Returns are normalized by single-task experts trained for 15M timesteps. Error bars indicate the maximum and minimum returns across three trials.

We compare Transformers trained by the "indirect" and $Q$-dependent update against the on-policy PPO agent from the POPGym codebase [27] using the best sequence model from the original

benchmark (a GRU RNN [98]). Figure 5 aggregates the results across all 27 tasks with returns scaled between a random policy and the best single-task result (of any sequence model) that appears in Morad et al. [99]. Single-task reference scores are given a budget of 15M timesteps, while our multi-task agents see a total of 100M across all 27 tasks and train on reward signals for approximately half of those timesteps. Both Transformer learning updates outperform PPO-GRU but are relatively comparable to each other. This may not be surprising because POPGym tasks have similar learning curves and returns bounded in $[-1, 1]$. However, many of these tasks have sharp short-term credit assignment where the advantage surface of an accurate value function would be steep. We expect the standard $\mathcal{L}_{\text{Actor}}$ update to be a significant improvement in online exploration, so it is encouraging that the offline-style $\mathcal{L}_{\text{Actor-Ind}}$ can exceed its performance after 15M total timesteps. Learning curves for each task can be found in Appendix C.

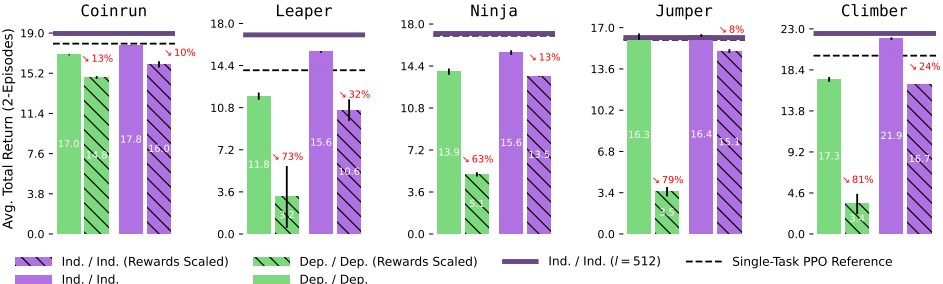

Figure 6: **Return Scales in Multi-Game Procgen.** We evaluate over 2 episodes in unseen levels after optimizing default or rescaled reward functions. Scores are averaged over two model checkpoints and converted to the default scale. Error bars indicate the difference between two 275M timestep trials.

**Multi-Game Procgen.** Procgen [28] creates 16 video games with an infinite variety of procedurally generated levels. If memory-based actor-critic learning is bottlenecked by the scale of returns across multiple tasks, we should be able to decrease performance by simply rescaling rewards. We begin by training multi-task agents across 2k levels of five games in easy mode. Solid bars in Figure 6 show the performance on unseen test levels, where the scale-resistant update offers a slight improvement.

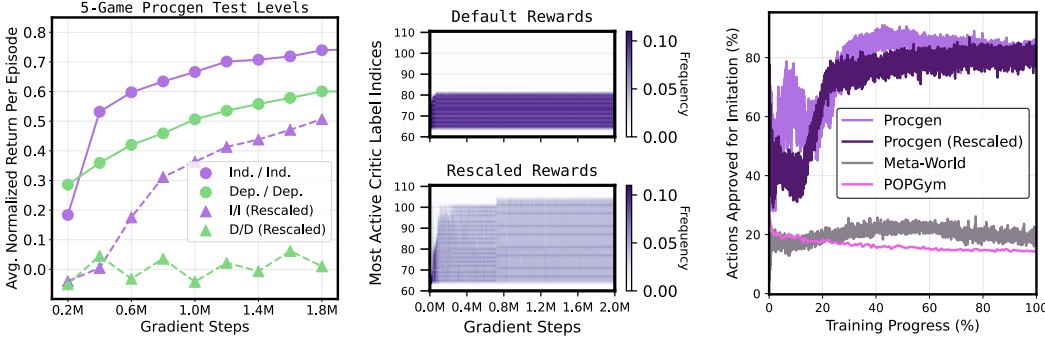

Figure 7: **Learning from Rescaled Returns. (Left)** Procgen test level returns normalized across games according to the standard benchmark scale [28]. **(Center)** Frequencies of the critic classification label index with the highest probability throughout training. We trim the y-axis to the portion of symlog [31] space that is relevant to Procgen. **(Right)** The value of $\mathcal{L}_{\text{Actor-Ind}}$ binary advantage weights create an estimate of the percentage of the replay buffer currently being imitated.

Our multi-task Procgen environment randomly selects a new game between resets, which does not account for how episode lengths depend on the agent's skill (Figure 14). We retrain new policies from scratch but scale the rewards of the easiest task (Coinrun) by $\times 100$ while dividing the rewards of the task the agent spends the most timesteps interacting with (Climber) by 10. We would expect the scale of $\mathcal{L}_{\text{Actor}}$ and $\mathcal{L}_{\text{Critic}}$ to be dominated by Coinrun so that optimization variance overshadows the objectives of the other environments even after learning in Coinrun converges. Hatched bars in Figure 6 show this is indeed the case, with methods recovering most of their original performance in Coinrun while failing to improve in the other four games. The "Ind. / Ind." update is far less impacted by the new return scales even though they decrease efficiency in terms of gradient steps (Figure 7 Left). Fig.

7 (Center) reveals a simple explanation where the broader range of returns approximately doubles the number of frequently used labels our critics need to learn to classify. Multi-task Transformer policies are commonly trained by imitation learning, and Fig. 7 (Right) highlights why $\mathcal{L}_{\text{Actor-Ind}}$ is so effective for training multi-task Transformer policies with online RL: the policy update reduces to IL on a dynamic percentage of the replay buffer, but the ability to ignore a fraction of the dataset automatically allows for self-improvement in a way that standard IL does not.

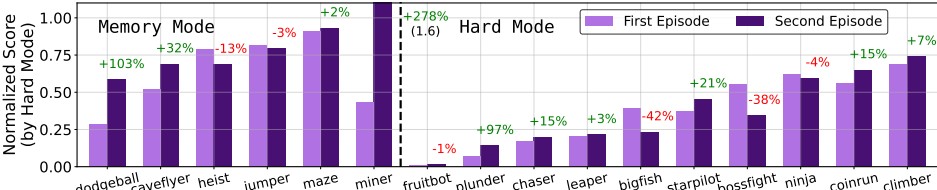

Figure 8: **Multi-Task Procgen in Memory-Hard Mode.** We measure policy performance across the two episodes of its adaptation window. Results are averaged over 30M frames in unseen test levels.

Inspired by the success of the "Ind. / Ind." learning update in the easy distribution of 5 games, we train a larger 24M parameter Transformer with a context length of $l = 768$ frames on all 16 Procgen games simultaneously. We default to the standard "hard" level distribution but increase the difficulty to the (rarely attempted) "memory" mode in the 6 games it is available. Results after 2.7B frames are highlighted in Figure 8. The agent does improve its performance on the second attempt of unseen levels — particularly in memory games where partial observability makes adaptation most useful.

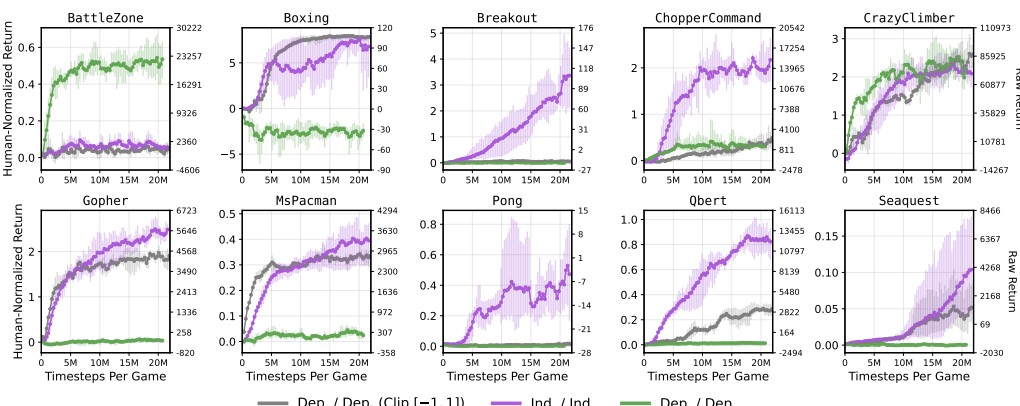

Figure 9: **Multi-Game Atari Without Reward Clipping.** We train a single policy on 10 games simultaneously. Results are plotted relative to human performance (left axis) and the raw scale of returns (right axis). Error bars denote the maximum and minimum over four random trials.

**Multi-Game Atari.** If RL generalization problems lie on axes between meta-RL and MTRL (Section 2), the ALE [29] would represent the multi-task extreme. Super-human performance on Atari does not require adaptation or memory but leads to per-game returns separated by orders of magnitude [100], which motivates the infamous trick of clipping rewards in $[-1, 1]$ [101]; clipping stabilizes learning but misaligns agents' training objective from the raw game score used for evaluation. PopArt [25] showed that multi-task training on unclipped rewards is possible when we split critic networks into game-specific output heads with independently normalized targets. Figure 9 revisits this experiment on a smaller scale. However, rather than normalizing a multi-task architecture, we train a single critic on 10 games with unclipped rewards and let our shared Transformer recover state and task labels implicitly from context sequences (of RGB images). In other words, we are treating the widely studied MTRL Atari setting as a special case of meta-learning in order to isolate the challenge of optimizing returns on different scales. Results are shown in Figure 9. The scale-resistant update leads to dramatic improvements in $8/10$ games; the default $Q$-dependent update performs best in the two games where returns at random initialization are significant outliers. The standard approach of clipping rewards in $[-1, 1]$ (grey) helps mask the multi-task optimization challenge created by scale-dependent learning updates.

**BabyAI.** BabyAI is a suite of partially observed gridworld tasks with simple language instructions [30]. We create a hybrid multi-task meta-RL problem with the potential for task generalization by randomly generating a train/test split over $68$ of BabyAI's task configurations as provided by the Minigrid simulator [102]. Our agents observe a text description of the their task and have two attempts to adapt to a procedurally generated level layout. Learning curves for all $68$ tasks are provided in Appendix C. Figure 10 summarizes our results in the BabyAI domain and reveals another case where scale-independent updates outperform scalar regression. Figure 11 evaluates BabyAI as a multi-episodic adaptation problem by comparing the returns of the first and second attempt in a new level layout. Despite BabyAI's partial observabilty, only a small set of tasks benefit from in-context adaptation over a context length $l = 512$. Transformers are clearly capable of learning effective recall over sequences of this length via RL [56, 57], and we have demonstrated that scale-invariant updates can unlock significant improvement in terms of multi-task training. Therefore we argue that the next step is to train this agent in a domain with sufficient task diversity to evaluate meta-learning over $k > 2$ episodes. XLand Minigrid [103] is a concurrent effort to augment gridworlds with millions of unique tasks and may be a promising domain for future study.

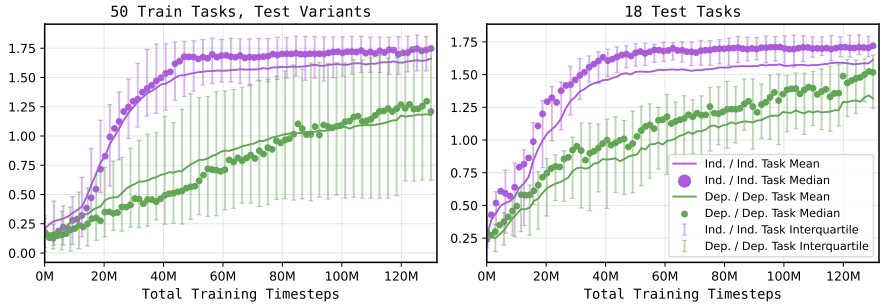

Figure 10: **Multli-Task BabyAI.** Results are the average over four training seeds and plotted according to the median, mean, and interquartile range over the task set.

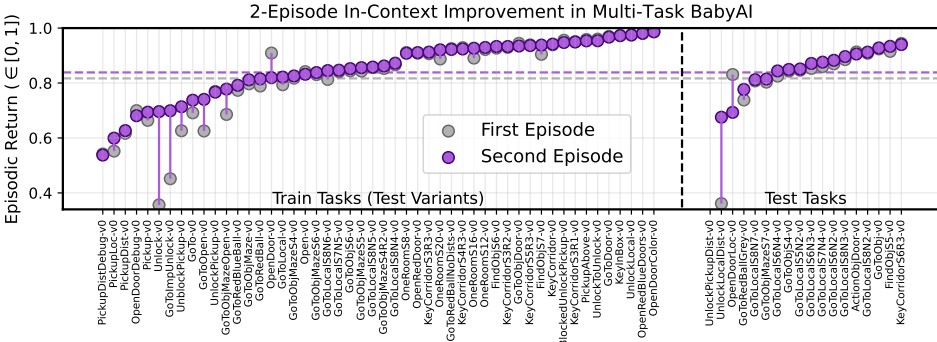

Figure 11: **In-Context BabyAI.** We measure the average return of "Ind. / Ind." agents by attempt in unseen tasks and/or layouts (variants). Results are arbitrarily sorted in order of increasing second-episode return and are the average of four training seeds and 10M evaluation timesteps.

## 5 Conclusion

Memory-based RL methods are already capable of long-term recall and test-time adaptation to variations of familiar tasks. However, optimization challenges from multi-task learning limit our ability to expand to more general applications. We attribute some, but not all, of this difficulty to the unpredictable way our agents balance the optimization of each task according to their current performance. We show that this bottleneck can be removed without explicit task labels or additional assumptions by simply replacing both the actor and critic objectives with classification terms that do not depend on the current scale of returns. Experiments across meta-learning, generalization, and long-term memory benchmarks find that this update improves multi-task performance without sacrificing sample efficiency. More broadly, these results justify a uniquely accessible approach to adaptive RL. Training a single Transformer on arbitrary recycled data with two loss functions that resemble supervised learning is a simple way to unlock the self-improvement of online RL while sharing intuition and technical details with widely familiar topics like sequence modeling.

## Acknowledgments and Disclosure of Funding

This work was partially supported by the National Science Foundation (EFRI-2318065), the Office of Naval Research (N00014-24-1-2550), the DARPA TIAMAT program (HR0011-24-9-0428), JP Morgan, and Sony.

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

# A Implementation Details

Code for the agent and multi-task environments used in our experiments is available on GitHub at UT-Austin-RPL/amago.

**Base RL Details.** We focus on evaluating changes to the training objective of long-term memory policies trained by off-policy actor-critic updates. Our implementation makes use of many orthogonal technical details from AMAGO [57]. In the context of this work, the most important detail is the use of a global (task-agnostic) PopArt [25] layer to normalize $Q$-values *of the "dependent" update* ($\mathcal{L}_{\text{Actor}}, \mathcal{L}_{\text{Critic}}$), which brings loss functions to a predictable absolute scale but does not change the fact that each task has different relative values. Trying to compare learning updates while skipping this technique would make the $Q$-dependent baselines far too reliant on accurate grid-searches of optimization hyperparameters across domains. Table 1 provides the list of hyperparameters used in our main experiments. All of the results in this paper were completed on NVIDIA A5000 GPUs. We train each agent on one GPU whenever possible but add a second GPU for Procgen Memory-Hard (Figure 8) where model size and context length use all available memory.

| | Meta-World | POPGym | Procgen Easy | Procgen Memory-Hard | Atari | BabyAI |
|---|---|---|---|---|---|---|
| Learning Rate | 5e-4 | 1e-4 | 1e-4 | 1e-4 | 1e-4 | 1e-4 |
| Batch Size (In Sequences) | 24 | 24 | 24 | 16 | 32 | 24 |
| Replay Buffer Size (in Timesteps) | 8M | 15,000 Full Trajectories | 18M | 80M | 8M | 10M |
| L2 Penalty | 1e-4 | 1e-4 | 5e-3 | 5e-3 | 5e-3 | 1e-3 |
| Grad Clip (Norm) | 2 | 1 | 1 | 1 | 1 | 1 |
| Learning Update s | | | [0.1, .9, .95, .97, .99, .995, .999] | | | |
| Rollout $\gamma$ | | | .999 | | | |
| Critic Ensemble Size | | | 4 | | | |
| Target Network Update $\tau$ | | | .003 | | | |

Table 1: **Learning Hyperparameter Details.**

**Model Architectures.** All of our experiments use causal Transformers trained with the AdamW optimizer [104]. We apply Normformer [105] and $\sigma$Reparam [106] with Leaky ReLU activations to all architectures. These details are intended to stabilize the policy against long training runs of millions of gradient steps but are applicable to any sequence modeling problem; our results did not require any changes that are specific to RL. We use fixed (sinusoidal) position embeddings [26]. Table 2 lists Transformer architectural details for each of our main experiments. The "timestep encoder" maps the potentially multi-modal meta-RL inputs of observations, actions, rewards, and reset signals to a fixed-size vector (Figure 2). State-based domains (Meta-World and POPGym) can concatenate and merge this information with a small feed-forward network. In pixel-based environments, we first embed the image observation with a CNN. We evaluated both the small "Nature" architecture [101] and the residual IMPALA CNN [107]. Our results default to IMPALA with additional Group Normalization [108, 65]. We apply the random pad and crop data augmentation from DrQV2 [109] to Procgen and Atari experiments. Image features extracted by the CNN are normalized before being added to action, reward, and terminal data. For consistency with the rest of the architecture we use LayerNorm [110]. Preliminary experiments found many of these vision details to be flexible.

| | Meta-World | POPGym | Procgen Easy | Procgen Memory-Hard | Atari | BabyAI |
|---|---|---|---|---|---|---|
| Timestep Encoder | FF (512, 256) | FF (512, 200) | IMPALA CNN [16, 32, 32] block depths | IMPALA CNN [20, 36, 64] block depths | IMPALA CNN [16, 32, 32] block depths | Multi-Modal CNN (Grid), RNN (Language), FF (Other) |
| Transformer Dim. | 320 | 256 | 512 | 512 | 256 | 256 |
| Transformer FF Dim. | 1280 | 1024 | 2048 | 2048 | 1024 | 1024 |
| Transformer Layers | 3 | 3 | 3 | 6 | 3 | 4 |
| Transformer Heads | 8 | 8 | 8 | 8 | 8 | 8 |
| Context Length (Timesteps) | 256 | 600 | 128 | 768 | 64 | 512 |
| Actor and Critic MLPs | (256, 256) | (256, 256) | (256, 256) | (256, 256) | (256, 256) | (256, 256) |
| Critic Output Bins (Ind.) | 128 | 64 | 128 | 128 | 128 | 32 |

Table 2: **Agent Architecture Details.**

**Value Classification Details.** We convert value regression (Equation 1) to classification (Equation 3) by creating labels for $B$ return bins $\mathbf{b} = [b_0, b_1, \ldots, b_B]$. Bins are typically spaced at fix intervals between pre-defined upper and lower bounds on the return ($b_0 = R_{\text{low}}, b_B = R_{\text{high}}$). The critic network ($Q_B$) outputs (softmax) probabilities over these bins, and its value prediction can be recovered by $Q(h_t, a_t) = Q_B(h_t, a_t)^{\mathsf{T}}\mathbf{b}$. A scalar temporal difference target $y$ is mapped to a classification label by $\text{twohot}_B : \mathbb{R} \to [0, 1]^B$ — a function that outputs zeroes everywhere aside

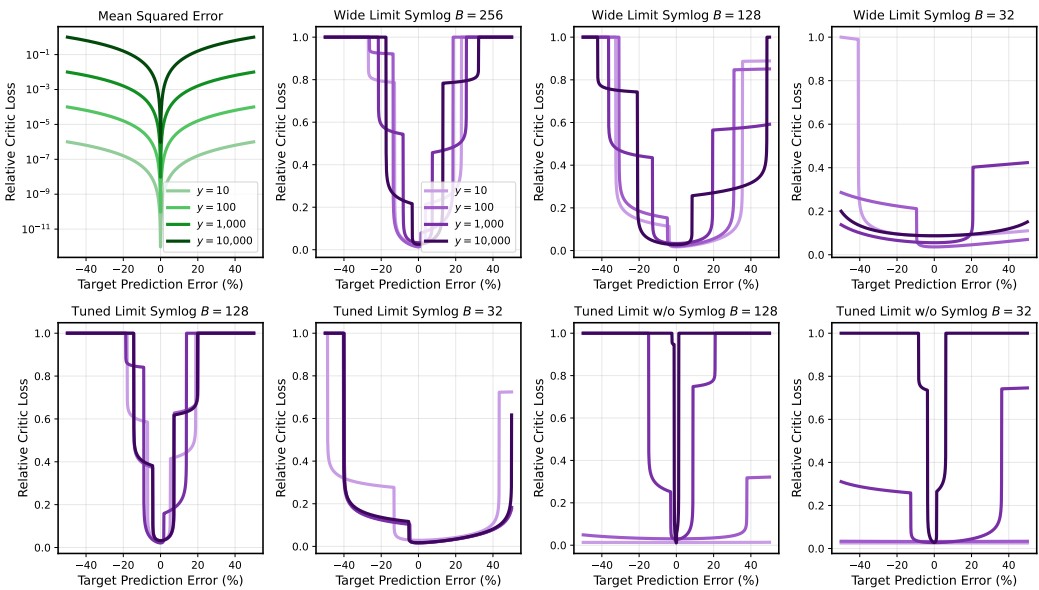

Figure 12: **Scale-Resistant Regression.** We plot the critic loss as a function of the *relative* prediction error of the value target ($y$) across several orders of magnitude. Standard MSE (top left) assigns significantly different weights to the same relative inaccuracy across tasks according to the absolute value of their current targets. Converting regression to classification can map a wide range of values to a similar loss, but this effect can benefit from tuning. We show 7 different strategies. "Wide Limit" refers to the DreamerV3 approach of setting extreme $(R_{\text{low}}, R_{\text{high}})$ defaults that do not need tuning. "Tuned Limit" bins set the upper and lower bound close to the expected range for the domain (in this example: $[-100, 15, 000]$).

from two consecutive indices corresponding to the closest bin below $y$ ($b_{i-1}$) and above $y$ ($b_i$), which are filled based on their distance from $y$:

$$\text{twohot}_B(y)[i-1] = \frac{y - b_{i-1}}{b_i - b_{i-1}} \quad (5) \qquad \text{twohot}_B(y)[i] = \frac{b_i - y}{b_i - b_{i-1}} \quad (6)$$

Therefore there are two main settings to configure for each experiment: 1) the bin limits $[R_{\text{low}}, R_{\text{high}}]$, and 2) the number of (evenly spaced) bins between the limits, $B$. We can often set bin limits using prior knowledge of the maximum and minimum return in a given domain, but it can be challenging to establish correct settings for a new application or reward function. The number of bins $B$ defines the "resolution" of the critic network. Using fewer bins puts more emphasis on outputting correct label weights while adding more bins shifts focus to outputting correct label indices. This trade-off is interesting because it is somewhat unusual for the label space of a classification problem to be a tunable hyperparameter; for example, the dataset determines the number of labels in object recognition or language modeling. Figure 13 provides two single-task examples where lower label counts are more sample efficient, and Figure 7 in the main text supports a similar conclusion in multi-task Procgen. These settings can be easily adjusted in our open-source code, but their trade-offs are underexplored in the results of this work. Instead, we focus on reducing tuning by following details in DreamerV3 [31]. We set $[R_{\text{low}}, R_{\text{high}}]$ extremely wide such that they capture the full range of every experiment and do not need to be tuned. We transform $y$ with "symlog" before label creation: $\text{twohot}_B(y) \leftarrow \text{twohot}_B(\text{symlog(y)})$. Scalar values are recovered from bin probabilities by inverting this transformation with "symexp": $Q(h_t, a_t) = \text{symexp}(Q_B(h_t, a_t)^\mathsf{T}\mathbf{b})$, where:

$$\text{symlog}(y) = \text{sign}(y)\ln(|y| + 1) \quad (7) \qquad \text{symexp}(b) = \text{sign}(b)(\exp(|b|) - 1) \quad (8)$$

Bins are spaced evenly between $b_0 = \text{symlog}(R_{\text{low}})$ and $b_B = \text{symlog}(R_{\text{high}})$. This symlog trick skews labels to give more resolution to smaller return values while retaining our ability to represent extreme outliers at a lower resolution. In general, we find that setting a wide limit $(R_{\text{low}}, R_{\text{high}})$,

increasing the label count $B$, and compressing labels with symlog/symexp is the safest way to achieve scale-invariance across domains (Section 3). An example is illustrated by Figure 12. We always use symlog and wide limits of $(R_{\text{low}} = -1e5, R_{\text{high}} = 1e5)$, and only tune the bin count $B$ (Table 2). However, it is likely that sample efficiency can be improved by tuning bin counts, limits, and the use of the symlog transform for each individual domain.

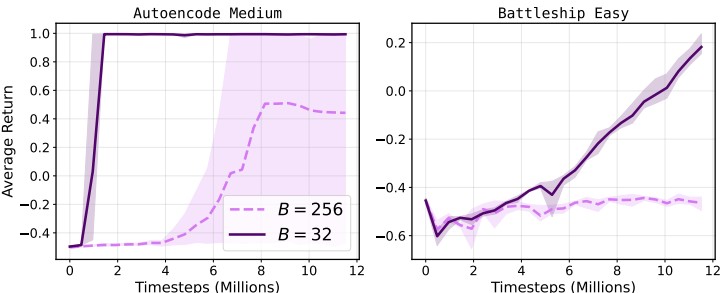

Figure 13: **Bin Counts in POPGym.** We compare two label space sizes ($B$) with the same $R_{\text{low}}$ and $R_{\text{high}}$ limits on two single-task POPGym environments [27].

## B    Additional Environment Details

**Multi-Task POPGym.**    The environment samples uniformly between 27 POPGym [27] tasks with discrete actions where both the action space and observation space have dimension $< 30$. A full list is provided as part of Figure 15. The cutoff of dimension 30 is arbitrary but some limit is necessary because POPGym has several tasks with unusually large action spaces. We are then able to create unified input and output spaces by padding observations and actions to dimensions of 26. If the agent selects an action index that is not valid for the current task, the environment samples a valid action uniformly at random. An additional observation feature indicates whether the previous action was valid. The one-shot evaluation setting is implemented as in E-RL$^2$ [68], where rewards for the exploration episode are inputs to the policy but are not allowed to become true reward outputs that are optimized by RL updates.

**Multi-Game Procgen.**    We merge individual Procgen [28] games into a multi-task environment by randomly sampling a new game between meta-rollouts. The same procedurally generated level is then played twice before a new game and level are sampled. In order to allow policies without long-term memory to form accurate value estimates, we draw a small black box in the top left corner of the screen throughout the second episode. We enforce a maximum rollout length (across both episodes) of $H = 2500$ for the 5-game easy mode (Figure 6) and $H = 5000$ for the memory-hard mode (Figure 8). In memory-hard mode, we sample levels from the "memory" distribution when available and "hard" otherwise. The memory distribution extends the hard difficulty by introducing partial observability. Memory mode was not evaluated in the original Procgen results and has been rarely used since. Therefore the results in Figure 8 are normalized according to the established scale for hard mode. With the exception of the "Miner" game, these upper bound scores transfer to memory mode as it does not adjust the problem difficulty but does hide information about the layout of the level.



One detail is that the random game between resets does not account for the length of each rollout in terms of timesteps or the way episode length increases or decreases as the agent improves. Our multi-task Procgen experiments create imbalanced datasets where each game makes up an uneven amount of incoming training data (Figure 14). We consider this to be an interesting additional challenge that off-policy methods with large replay buffers are well suited to address.

Figure 14: **Imbalanced Procgen Datasets.** We measure the inflow of experience to the replay buffer by game. Climber accounts for much of our early training data but the buffer is rebalanced as policies improve.

**Atari.** We select 10 games that do not involve challenging exploration but that have human-level returns on several different orders of magnitude. The list of games is included in Figure 9. We use the standardized v5 variants of the Atari environments in Gymnasium [111]. In addition to removing reward clipping, we do not use frame stacking or greyscaling. Instead, agents can learn short-term state estimation and task identification from context sequences of RGB frames.

**BabyAI.** BabyAI provides the agent with a $7 \times 7$ partial view of its surroundings in a procedurally generated gridworld. Goals are communicated with relatively simple language instructions, which we represent as sequences from a vocabulary of 29 tokens. Each reset randomly samples a new goal type and gridworld generation strategy from 50 tasks in the Minigrid registry during training, and 18 tasks during testing. A full list of task names is included as part of Figure 11.

## C  Additional Results

Large figures displaying results from each task in our POPGym (Figure 15) and BabyAI (Figure 16) experiments are listed on the following two pages. These figures are followed by learning curves for each Meta-World ML45 task.

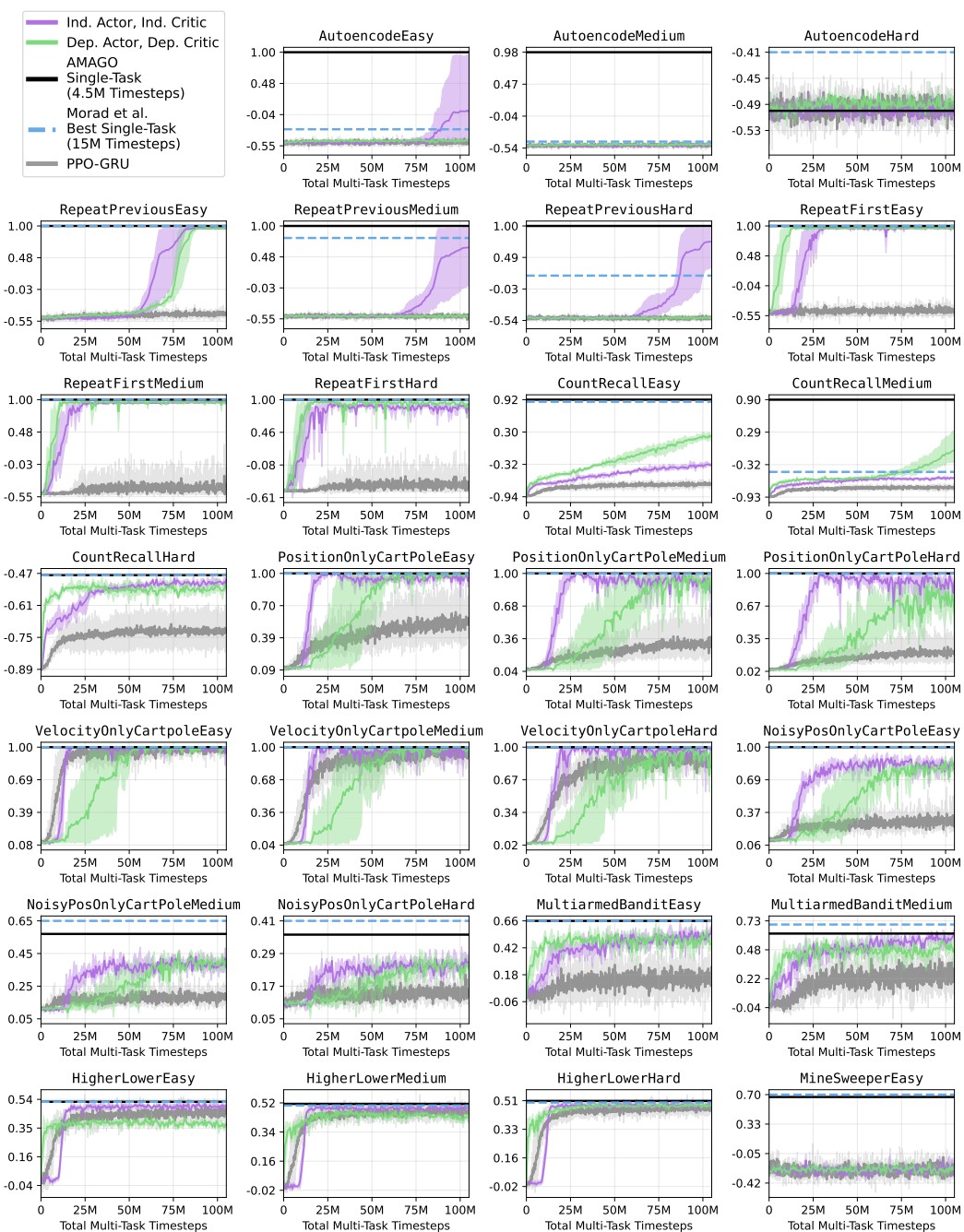

Figure 15: **One-Shot Multi-Task POPGym.** Plotted on the raw scale of returns in each environment. Error bars denote the maximum and minimum results over three trials. Single-Task reference scores at 15M timesteps indicate the best of 14 sequence model backbones trained by PPO in the results of Morad et al. [99]. AMAGO [57] single-task reference scores are taken at 4.5M timesteps, which is a safe upper-bound on the timesteps encountered for each task during multi-task training.

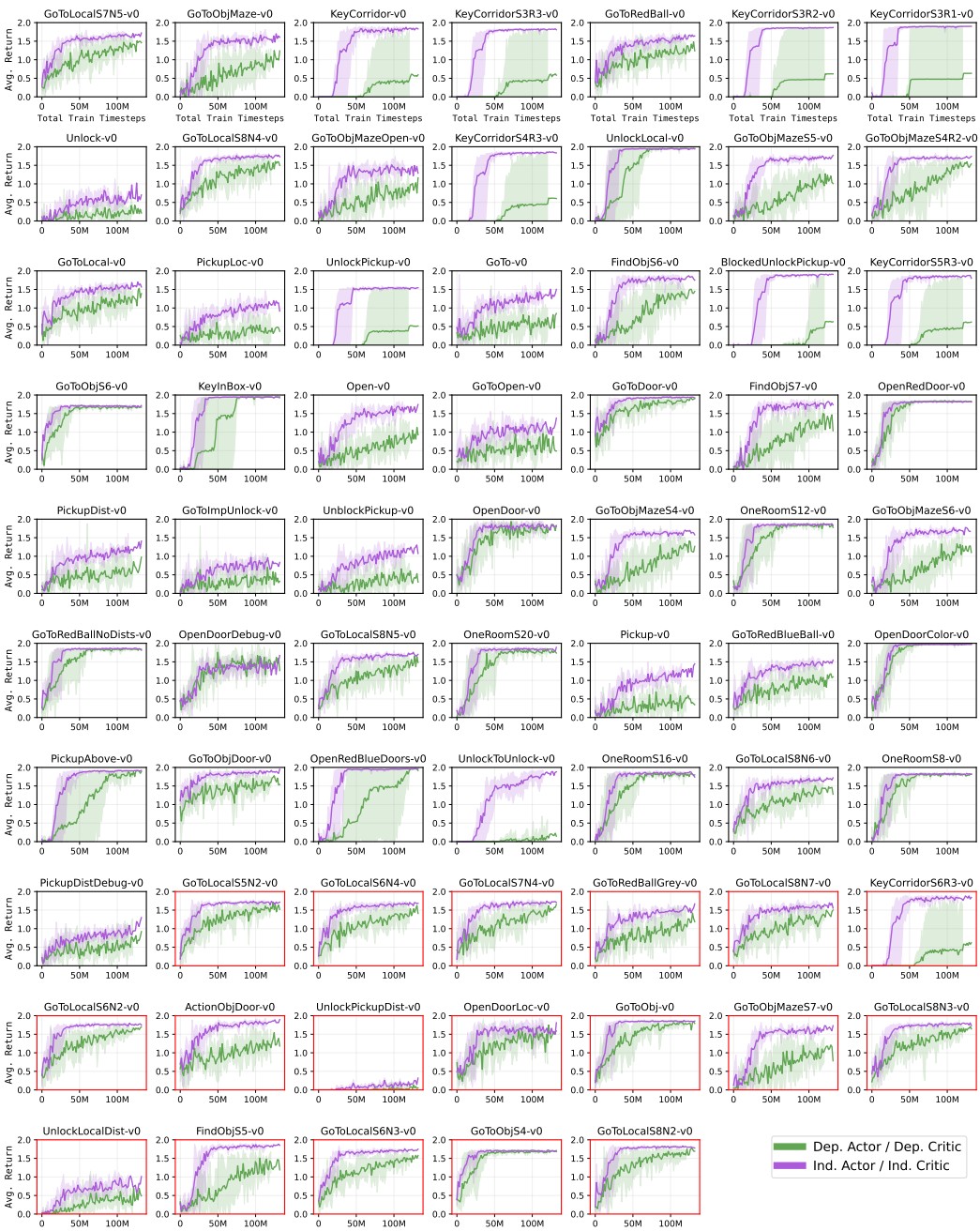

Figure 16: **Two-Episode Multi-Task BabyAI.** Held-out test tasks are highlighted in red. Error bars denote the maximum and minimum value of four random trials.

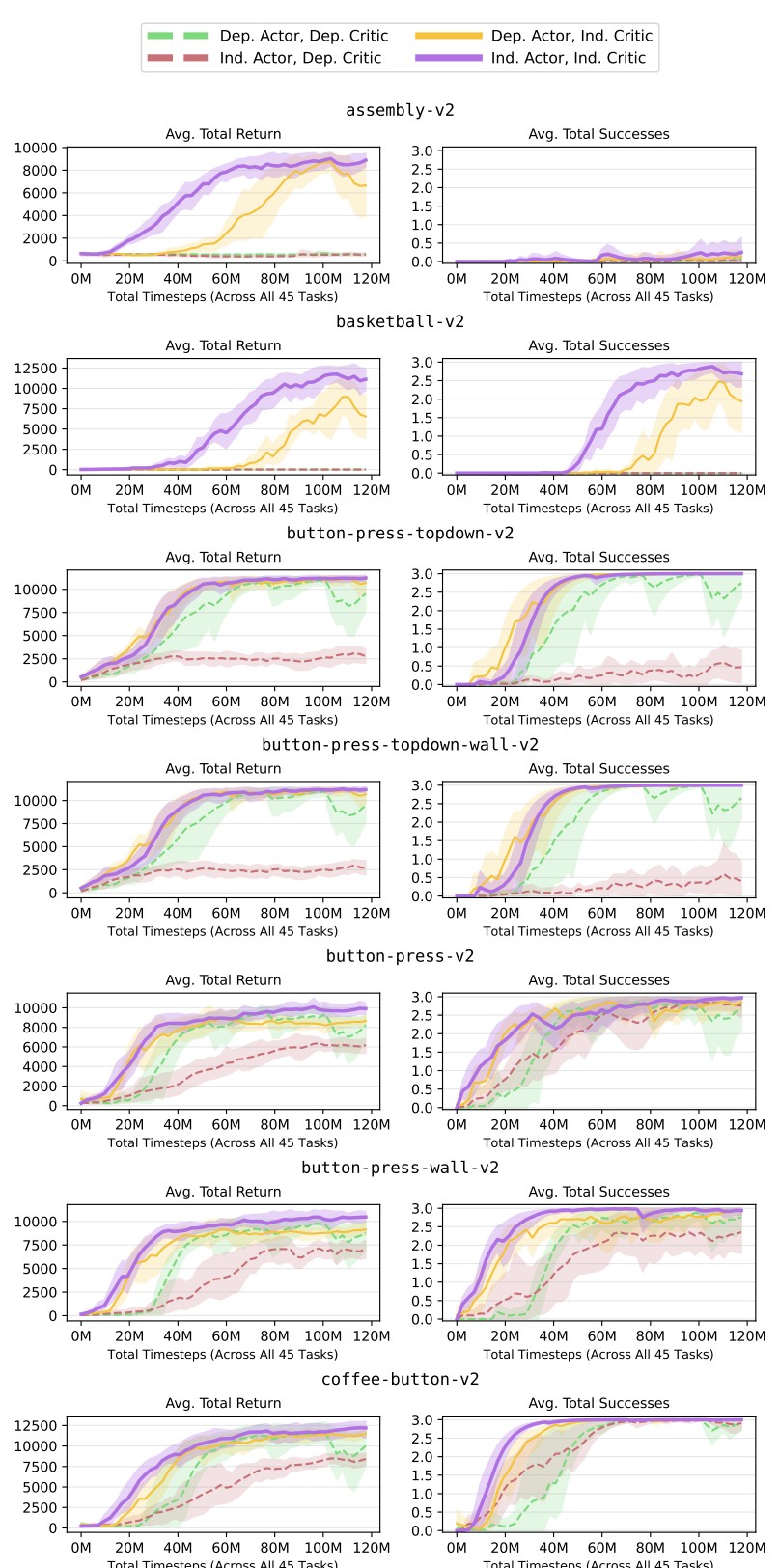

Figure 17: **Three-Episode Meta-World ML45 Learning Curves (Part 1/7)**

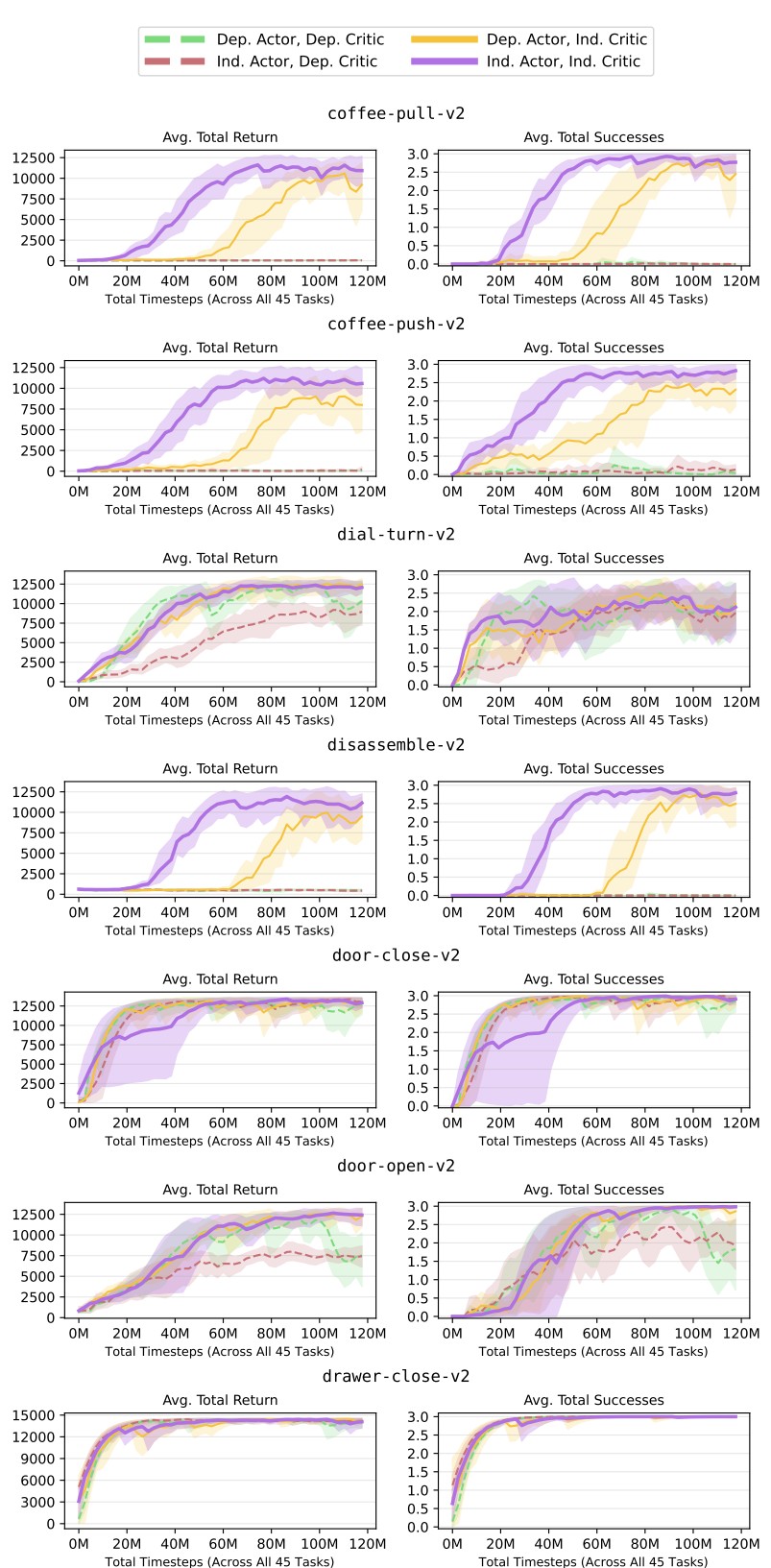

Figure 18: **Three-Episode Meta-World ML45 Learning Curves (Part 2/7)**

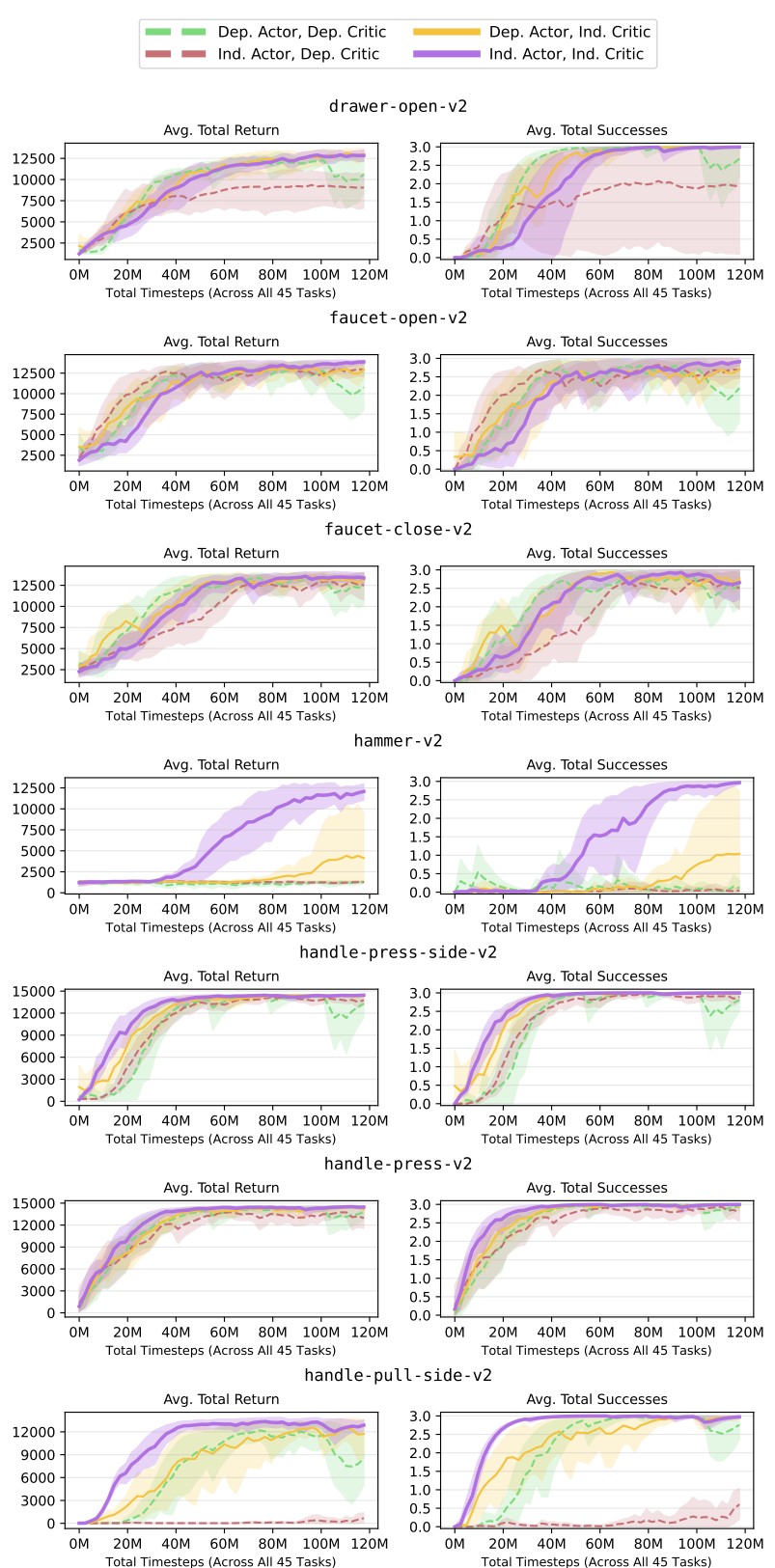

Figure 19: **Three-Episode Meta-World ML45 Learning Curves (Part 3/7)**

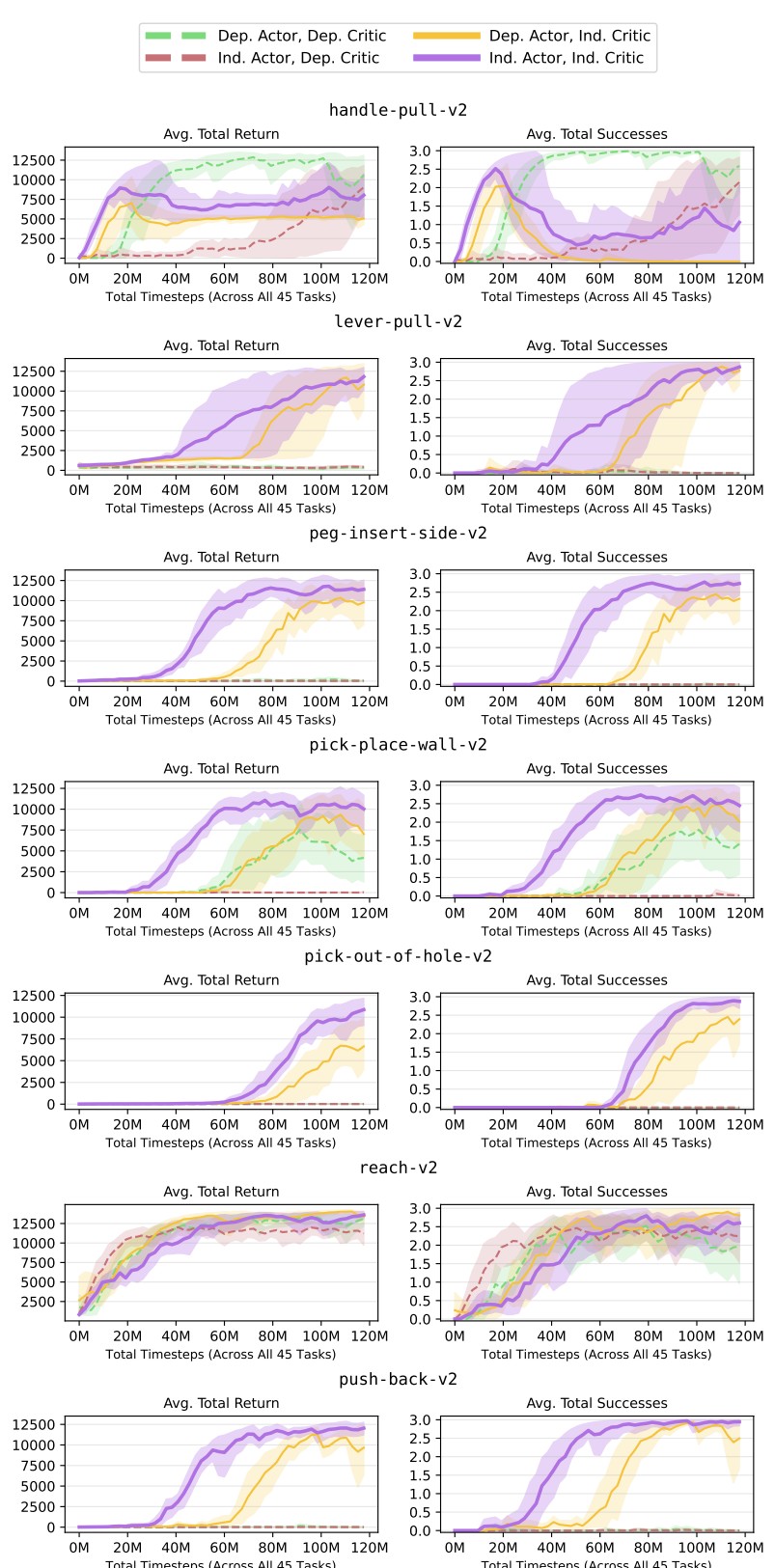

Figure 20: **Three-Episode Meta-World ML45 Learning Curves (Part 4/7)**

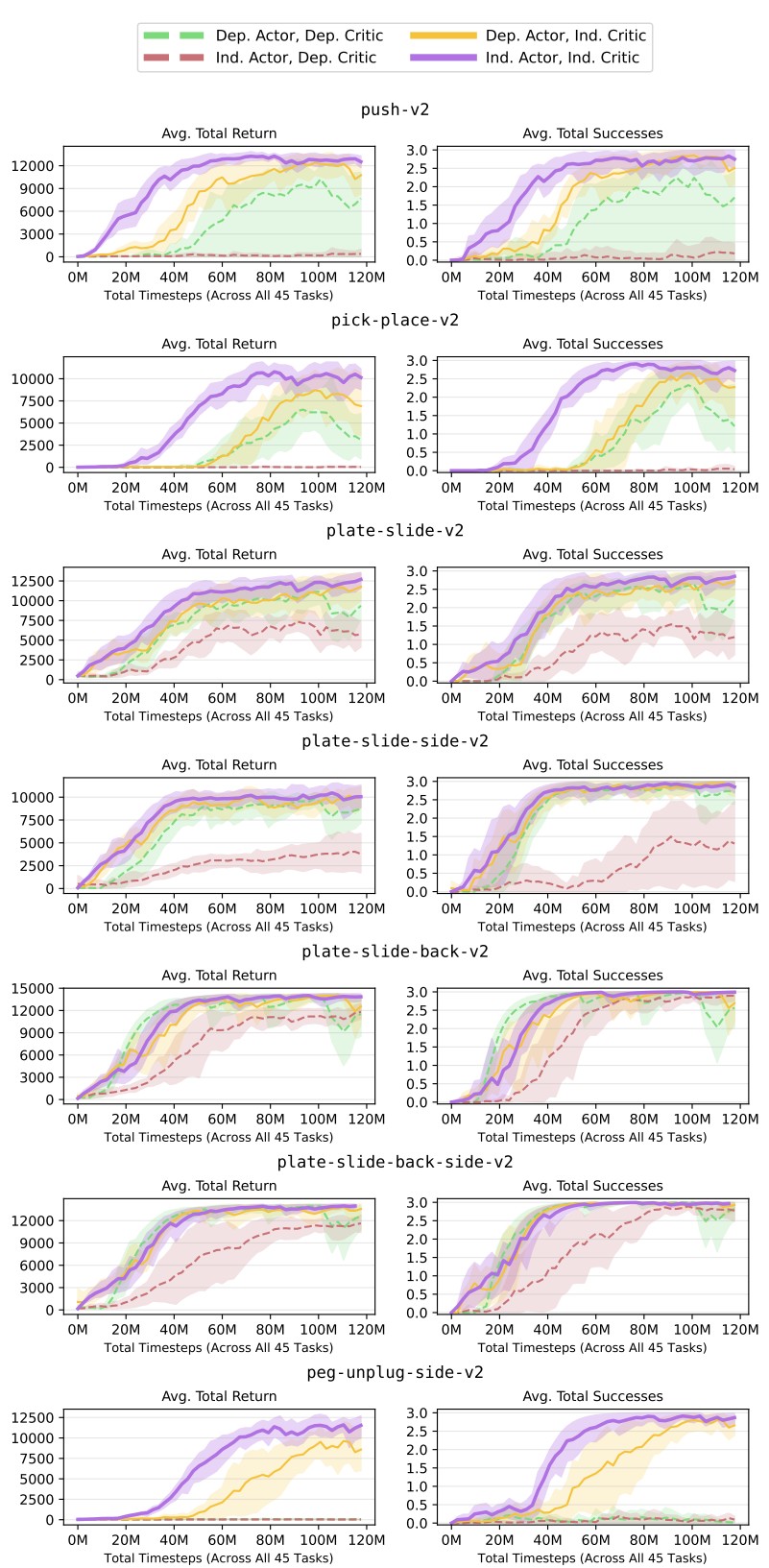

Figure 21: **Three-Episode Meta-World ML45 Learning Curves (Part 5/7)**

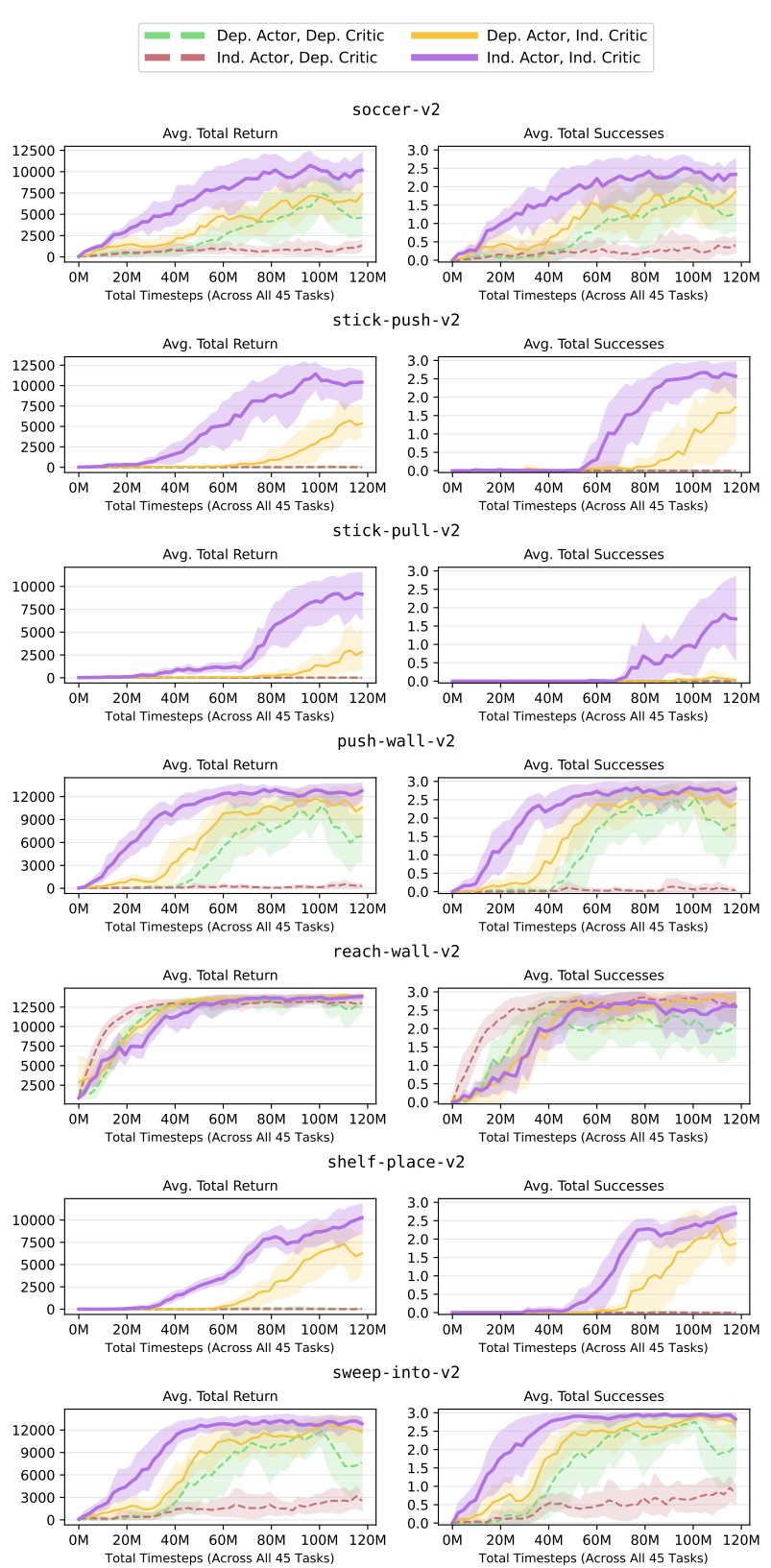

Figure 22: **Three-Episode Meta-World ML45 Learning Curves (Part 6/7)**

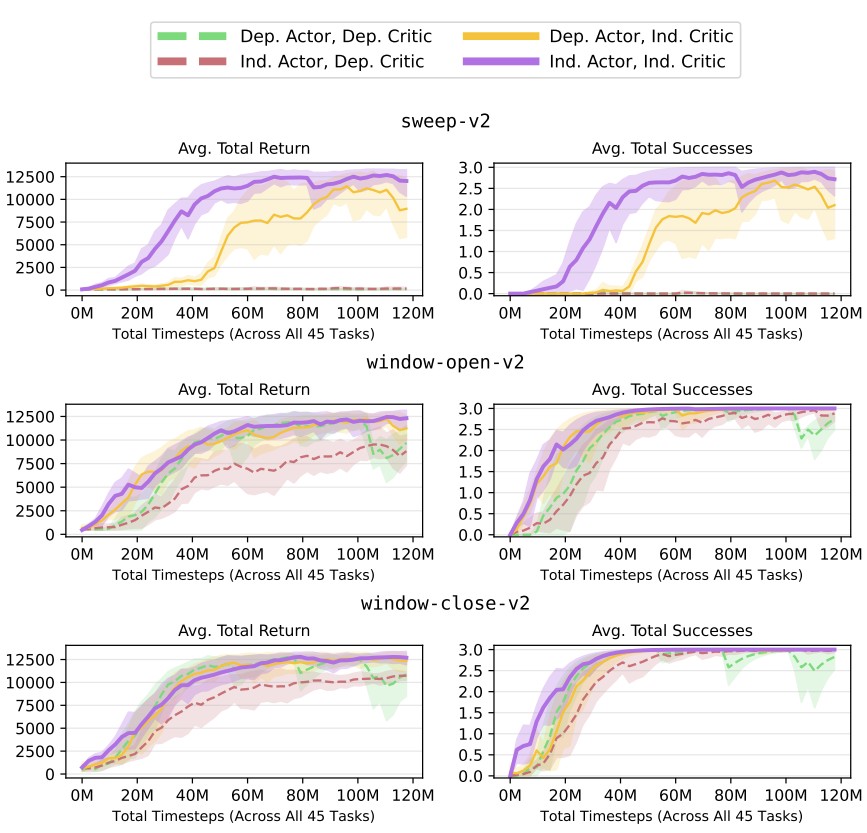

Figure 23: **Three-Episode Meta-World ML45 Learning Curves (Part 7/7)**

