# OpenReview forum: "AMAGO-2: Breaking the Multi-Task Barrier in Meta-Reinforcement Learning with Transformers"
_NeurIPS.cc/2024/Conference — NeurIPS 2024 poster_

### Official Review · Reviewer_TXBg · 2024-06-15

**Soundness:** 4
**Presentation:** 3
**Contribution:** 3
**Rating:** 7
**Confidence:** 4

**Summary:**

This paper proposes a method for training transformer-based policies for multi-task meta-RL settings, where the task distribution has multiple tasks each of which has parametric variation and the training and test sets both include all tasks. Focus is on developing a method that handles the scale variation in returns between different tasks. Strong empirical results.

**Strengths:**

- The method description is clear and well motivated.
- The empirical results are strong and clearly presented.

**Weaknesses:**

- Framing issue: The considered setting is interesting and a fine target for research. However, parametric variation is not the grand challenge in meta-RL and the paper, in my opinion, does not do enough to clearly delineate the subproblem it tackles.
    - The ultimate goal of meta-RL is training an adaptive agent that can learn any new task it is presented with, not just parametric variation of existing tasks. This paper does not explore generalization to unseen non-parametric task variants. This should be more clearly stated early in the paper.
    - The test set of Meta-World ML45 benchmark is mentioned in the experiments, and reasonably ignored. However, since ML45 is already mentioned in the abstract, it would be better to mention that the paper only considers the training set.
- Minor framing issue: Line 27 says meta-RL is about task identification at its core. While many meta-RL tasks can be solved via methods that reduce to identifying the task and then doing multi-task learning, meta-RL as a whole isn't limited to that. The paper adopts a broader viewpoint in the scaling beyond multi-task barrier paragraph. It would be better to update the introduction to match this perspective.

**Questions:**

- This is out of scope for the review, but I'm curious, what do you think is the best bet for a meta-RL benchmark that would enable investigating the non-parametric generalization ml45 fails to provide?

**Limitations:**

Limitations are discussed briefly in the conclusion. Especially limitations concerning the non-parametric generalization mentioned in weaknesses should be discussed more extensively.

---

> ### Author Rebuttal · Authors · 2024-08-07
>
> Thank you for the review and for the constructive comments on limitations and writing. We will discuss your suggestions below.
>
> > Framing issue: The considered setting is interesting and a fine target for research. However, parametric variation is not the grand challenge in meta-RL and the paper, in my opinion, does not do enough to clearly delineate the subproblem it tackles.
>
> Yes, we fully agree that the goal should be adaptation to new tasks. Our perspective is that non-parametric variation will be possible if we can scale up our training set to keep that variation within distribution. This will likely take far more tasks than are available in benchmarks like Meta-World, and we generally think that attempts to measure or improve non-parametric variation on Meta-World, Atari, Procgen, and similar benchmarks can risk drawing conclusions from too small of a training set. The challenge is that we still struggle to overfit these small task sets and mostly leave this issue to be tackled by MTRL techniques where solutions might not necessarily be transferable to larger benchmarks. A first step might be to address the key MTRL challenges while maintaining the label-free meta-RL perspective. We will add more discussion to the paragraph on lines 42-55 of the introduction.
>
> >The test set of Meta-World ML45 benchmark is mentioned in the experiments, and reasonably ignored. However, since ML45 is already mentioned in the abstract, it would be better to mention that the paper only considers the training set.
>
> Yes, we can do that. We are not aware of any results that demonstrate strong transfer to the test set without introducing some restriction on the environment or policy to regularize training or without adding additional knowledge like language, goal states, or demonstrations.
>
> The reviewer is clearly familiar with Meta-World, but for any other readers who may not be, the train set of 45 tasks can still serve as a reasonable test of parametric variation because the environments are carefully designed to make these changes unobservable.
>
> > Minor framing issue: Line 27 says meta-RL is about task identification at its core … The paper adopts a broader viewpoint in the scaling beyond multi-task barrier paragraph. It would be better to update the introduction to match this perspective.
>
> We will rephrase to align this with the more detailed explanation given in Section 2. This line was meant to be a fast summary of the meta-RL perspective where adaptation reduces to exploring the environment to infer missing details of the unobserved parameters of the current task, which lets us treat multi-task RL problems as a special case.
>
> > This is out of scope for the review, but I'm curious, what do you think is the best bet for a meta-RL benchmark that would enable investigating the non-parametric generalization ml45 fails to provide?
>
> Meta-RL is facing a difficult benchmarking challenge at the moment, and it’s the reason our experiments need to drift into long-term memory and multi-task domains. A more affordable subset of XLand 2 [1] would probably be the best large-scale benchmark if it were accessible. There have been some recent efforts to create an open-source grid world version of this domain. One interesting direction for large-scale experiments (that is open-source) could be Gym Retro, which supports thousands of ALE-like games. Arcade games and platformers are a natural fit because they often give the player multiple lives/attempts before a reset. Meta-learning was the original goal of Retro, but this is obviously challenging and there hasn’t been much progress made.
>
> [1] *Human Timescale Adaptation in an Open-Ended Task Space*, Adaptive Agent Team, 2023

---

> > ### Comment · Reviewer_TXBg · 2024-08-11
> >
> > Thanks for the thoughtful response. I believe the paper would be a welcome contribution to meta-RL.

---

### Official Review · Reviewer_NjDj · 2024-07-08

**Soundness:** 3
**Presentation:** 3
**Contribution:** 3
**Rating:** 6
**Confidence:** 4

**Summary:**

In this paper, the authors investigate to utilize the Transformer architecture for multi-task RL tasks by giving the trajectories from multiple previous episodes as input tokens. More specifically and technically, they addressed the issue of the imbalanced rewards from different task (e.g., For Atari games, some games can give very high valued rewards while it is not for other games). It has been hypothetically resolved by utilizing additional knowledge such as the task label or normalizing rewards. They tried to address this without the additional knowledge for the task different from the previous works. Through this, they showed their modeling is effectively working well for diverse multi-task environments, especially when the reward distribution for each task is very different.

**Strengths:**

- Their problem setting is well positioned. As they pointed out, lots of meta-RL environments are mainly designed to evaluate that the agent can work when there are the small different in the tasks such as different positions in a single task, but we should aim to more extensive range of tasks to build the generalist agent. To do that, the imbalanced reward issue is importantly addressed even it has been handled with additional knowledge such as task label.
- This paper is well written for their motivation and proposal. Their writting for their motivation and proposal is very clear and good enough to understand for me.
- They reported strong experimental analyses. They did experiments on diverse environments such as Meta-World, POPGym, Procgen, Multi-Game Atari and Gym Retro for their modeling by comparing it to the ablation versions with and without some components of their modeling. Those results show that the strength of their modeling (better handling the imbalanced rewards without task label). Additionally, they also analyzed the in-context learning performance on Procgen environment.

**Weaknesses:**

- Some experimental analyses are hard to understand. I will ask it in Questions section.

**Questions:**

- Questions on the experimental results
	- For Multi-Game Procgen experimental analysis, especially the result in Figure 6 right, I am not sure that I understood correctly. You analyzed as "the policy update reduces to IL on a dynamic percentage of the replay buffer, but the ability to ignore a fraction of the dataset automatically allows for self-improvement in a way that standard IL does not.". What I understood from this analysis is that your agent can reduce to train the policy by imitating the given action, then the y-axis value of the graph should be increased as training progress goes on. But it is different for different environments, which is that what you mentioned as the "ignore a fraction of the dataset automatically"?
	- I didn't understand clearly the Gym Retro results. Why did you do this experiment? As I understood, it is to evaluate the larger scale of difference between tasks, why Gym Retro gives the setting for this testing? How did you set to test this? What means the SuperMario1 and 3? What means SuperMario2japan and SuperMarioWorld? How could I analyze the experimental results?
- For Multi-game Atari, can you compare this results with the single task agent results such as DreamerV2 [1] or state-of-the-art results on this environment? I want to compare the relative performance of your model with the single task agent like shown in Figure 5.
- For results in Figure 7, can you test it with diverse length of context? I am interested what happens if very large number of context is given.

[1] Hafner, Danijar, et al. "Mastering atari with discrete world models." arXiv preprint arXiv:2010.02193 (2020).

**Limitations:**

Toward generalist agent, their investigation can be seen as being limited in the homogenous environment setting. For example, extending it to train a single agent for multiple environments where action spaces are different as did in [1]. However, it can be seen as over their scope, I didn't think it is a critical limitation of their study.

[1] Reed, Scott, et al. "A generalist agent." arXiv preprint arXiv:2205.06175 (2022).

---

> ### Author Rebuttal · Authors · 2024-08-07
>
> Thank you for the review and comments on the experimental results. The evaluation domains in our work are computationally expensive, so it was not possible for us to fully address your questions with finalized results during the rebuttal window. We have begun work on these experiments and will update the paper when the results are complete.
>
>
> > For Multi-Game Procgen experimental analysis, especially the result in Figure 6 right, I am not sure that I understood correctly.
>
>
> The actor loss is performing supervised learning on a filtered subset of (obs, action) pairs from the replay buffer that are estimated to have a positive advantage by the critics. This update would be equivalent to imitation learning if the filter were to give a weight of one to every example, and we are relying on an accurate critic to mask low quality actions from the loss. Figure 6 right measures the percentage of each batch that passes the filter (A(s, a) > 0) and is used to compute the supervised loss. When the critic network is initialized it approves approximately half of the actions. As training continues, the curve changes according to a combination of the rate of experience collection, environment difficulty, gradient steps per environment step, and the total size of the replay buffer.
>
> > For Multi-game Atari, can you compare this results with the single task agent results such as DreamerV2 [1] or state-of-the-art results on this environment? I want to compare the relative performance of your model with the single task agent like shown in Figure 5.
>
> Atari is a highly optimized domain and we are not claiming state-of-the-art results here by any means. Our goal with this experiment is to use Atari as an example of a standard multi-task setting without the multi-episode adaptation or partial observability of the preceding results. We are measuring whether the change in training objective is enough to make progress without the established Atari trick of clipping rewards in [-1, 1] and multi-task RL techniques that rely on per-task gradients or network outputs. We want to measure a direct ablation of the same basic method that does not add orthogonal details that have led to much of the sample efficiency gains on the ALE. We will work to aggregate the results of single-task versions of the method using the default scale-dependent update (in green) for additional context.
>
> > For results in Figure 7, can you test it with diverse length of context? I am interested what happens if very large number of context is given.
>
> The trade-off between context length and performance depends on the episode length of the environment. The high frame rate of pixel-based domains can create much longer episodes than typical toy meta-RL experiments. Ideally, the context would span the entire length of the 2-episode rollout, but computational constraints (GPU memory) prevent this. It is worth noting that although we are not using as long of a context as we would like, 768 images is still an unusually long input sequence when training online RL from scratch.
>
> > their investigation can be seen as being limited in the homogenous environment setting. For example, extending it to train a single agent for multiple environments where action spaces are different as did in [1] … I didn't think it is a critical limitation of their study.
>
> Yes we are padding the observation/action spaces to the same shape across environments when necessary. A variable-length token encoding like Gato would probably be the first step in attempting this kind of heterogeneous setting. We avoided this because it adds significant computational cost in terms of sequence length per timestep of policy memory, and we felt that the main point of return scale variance is best demonstrated in domains where the observation and actions are similar but the reward functions are different.
>
> > I didn't understand clearly the Gym Retro results. Why did you do this experiment?
>
> This experiment is meant to be an extension of the Atari setting where rewards are on very different scales. The difference between the games we’ve chosen here and Atari is that the Mario levels are similar enough where some kind of zero-shot transfer might be realistic. The SuperMario names refer to the titles of the Mario video game series that are supported by the Gym Retro environment. We are working to add more context to these results by including the other filtered imitation learning ablation (“Ind. Actor / Dep. Critic” - the red curve in Figure 3) as another reference. The "Dep. Actor / Dep. Critic" comparison is not possible here due to the action space of the environment. These runs began before the reviews were released but are expensive to collect and have not finished. They will be added to Figure 9.

---

> ### Comment · Reviewer_NjDj · 2024-08-11
> **Reply to the rebuttal**
>
> Thank you the authors for their rebuttal. They substantially addressed my concerns, so I maintain my score.

---

### Official Review · Reviewer_1L6X · 2024-07-10

**Soundness:** 3
**Presentation:** 3
**Contribution:** 3
**Rating:** 6
**Confidence:** 4

**Summary:**

This paper addresses the challenge of scaling meta-reinforcement learning (meta-RL) to handle multiple tasks without explicit task labels. It introduces a method where both the actor and critic objectives are converted to classification terms, decoupling optimization from the scale of returns. This approach builds upon recent Transformer-based meta-RL techniques. The effectiveness of this method is demonstrated through large-scale comparisons on several benchmarks, including Meta-World ML45, Multi-Game Procgen, and Multi-Task POPGym. The results show significant improvements in online multi-task adaptation and memory problems.

**Strengths:**

### S1. Novel Approach and Problem Identification

The paper addresses the critical issue of imbalanced training losses in multi-task RL, which often arise due to uneven return scales across tasks. By converting the actor and critic objectives to classification terms, the method effectively decouples optimization from return scales, which is a novel and practical approach.

### S2. Empirical Validation

The method is validated through comprehensive experiments on multiple benchmarks, demonstrating significant improvements in task adaptation and memory problems. The results are clearly presented, with detailed comparisons to existing methods, highlighting the advantages of the proposed approach.

### S3. Clarity of Presentation

The paper is well-structured, with clear and concise explanations of the proposed method. Figures and tables are effectively used to illustrate results, enhancing the clarity of the presentation.

**Weaknesses:**

### W1. Lack of Theoretical Motivation and Analysis

The paper focuses heavily on empirical results but lacks a thorough theoretical analysis of why the proposed method outperforms existing approaches. A deeper exploration of the theoretical foundations and a comparison with existing theoretical frameworks would strengthen the paper.

### W2. Limited Generalization Analysis

While the method is validated on several benchmarks, the paper would benefit from a broader analysis of its generalization capabilities. Additional experiments on more diverse datasets and environments would provide a more comprehensive assessment of the method's applicability. For example, meta-world to multi-game procgen.

### W3. Assumption of Return Scale Variability

Although the paper discusses that return scales across tasks can vary significantly, the proposed method does not explicitly address how to handle extreme variations in return scales beyond the classification transformation. A more detailed discussion on how the method can be adapted or modified for tasks with highly diverse return scales would be valuable.

**Questions:**

### Q1. Scalability

What are the computational requirements and scalability of the proposed method when applied to very large-scale environments and datasets? Are there any practical limitations or considerations?

**Limitations:**

The authors have acknowledged the limitations related to the variability of return scales and the focus on specific benchmarks. However, a more detailed discussion on potential negative societal impacts and how to mitigate them would be beneficial.

---

> ### Author Rebuttal · Authors · 2024-08-07
>
> Thank you for your review. We will try to address your questions and would be happy to continue the discussion.
>
> > By converting the actor and critic objectives to classification terms, the method effectively decouples optimization from return scales, which is a novel and practical approach.
>
> To clarify, the classification loss terms are an implementation detail that appears in recent work we have cited and discussed in Sections 2 and 3. We do not want to claim the objectives themselves are novel. Instead, we show that while these ideas may have various motivations and minor benefits in other settings, they provide a major improvement to a key problem in the area of meta-learning and generalization without task labels.
>
> > The paper focuses heavily on empirical results but lacks a thorough theoretical analysis of why the proposed method outperforms existing approaches.
>
> We would be open to suggestions about how we can expand our analysis. Prior works have studied the challenges of multi-task optimization (Section 2). We do not think our work adds to the understanding of this issue, but it does help us remove an extra challenge where RL methods unintentionally place the objective of each task on a different scale (Section 3). We have added a more thorough explanation of this effect in a new figure attached to the rebuttal material.
>
> > While the method is validated on several benchmarks, the paper would benefit from a broader analysis of its generalization capabilities … For example, meta-world to multi-game procgen.
>
> We agree that transfer between domains like meta-world → multi-game procgen would be very interesting. This kind of transfer between different observation/action spaces would require a specialized input/output format for multi-modal trajectories of different shapes. Gato [1] is a good example for how we might go about this, but their method stretches input sequence lengths in a way that makes it expensive to evaluate memory.
>
> > Although the paper discusses that return scales across tasks can vary significantly … A more detailed discussion on how the method can be adapted or modified for tasks with highly diverse return scales would be valuable.
>
> In our experience, the extreme limits of value classification can be managed by globally rescaling all the rewards to a more stable range. The more challenging issue is the way variations in task difficulty drive uneven changes in values even when the initial and optimal returns are fairly well-bounded. For example, Multi-task POPGym and Meta-World ML45 both have bounded returns, but the benefits of the classification-style loss are much more clear on ML45 than POPGym. Reducing the number of classification bins may help improve performance in tasks where returns rapidly shift from the lower bound to the upper bound. We have added a demonstration of this on a POPGym task in the extra rebuttal material.
>
> > What are the computational requirements and scalability of the proposed method when applied to very large-scale environments and datasets? Are there any practical limitations or considerations?
>
> The learning update itself scales with the forward/backward of a single sequence model policy. It is comparable to imitation learning because the relative cost of the extra critic term decreases as the size of the shared Transformer increases. It does not directly depend on the task count or dataset size (replay buffer). The classification-style critic loss can be slower than standard regression as we have increased the dimension of the critic networks’ outputs to B bins. Relative to typical online RL experiments, we are evaluating large policies (10-20M+ parameters) and large datasets (20M+ timesteps). All the experiments were conducted on NVIDIA A5000 GPUs. The model is trained on one GPU while an extra GPU can be used for data collection to reduce wall-clock time.
>
> Please see our global reply for a discussion of limitations.
>
> > a more detailed discussion on potential negative societal impacts and how to mitigate them would be beneficial
>
> Sequence-based RL on off-policy data is a flexible approach to memory, adaptation, and generalization settings. The aim of our work is to make this technique easier to use by making more complex multi-task benchmarks realistic for research. We do not think our method adds negative societal impacts beyond the existing risks of RL systems, but we would be happy to discuss this further if you have concerns.
>
>
> [1] *A Generalist Agent*, Reed et al., 2022

---

> > ### Comment · Reviewer_1L6X · 2024-08-11
> > **Acknowledgement**
> >
> > I appreciate the authors for the detailed response.
> > Most of my concerns are addressed and I think this is a good paper to be accepted, so I maintain my assessment.

---

### Official Review · Reviewer_Ypn5 · 2024-07-13

**Soundness:** 2
**Presentation:** 3
**Contribution:** 1
**Rating:** 3
**Confidence:** 5

**Summary:**

This paper studies multi-task reinforcement learning using a context-based Transformer policy without task labels. To address optimization difficulties caused by imbalanced losses across different tasks, it proposed replacing actor and critic losses with classification losses. Ablation studies on several benchmarks show that the proposed classification-based actor-critic losses outperform the original algorithm.

**Strengths:**

1. The paper is well-focused on the problem of balancing losses in multi-task RL. The proposed classification losses are well-motivated and make sense.
2. Diverse benchmarks are used to evaluate the method.

**Weaknesses:**

1. Multi-task RL using in-context adaptation without task labels is not new. Some recent works [1,2] learn in-context meta-RL policies that adapt to diverse tasks, rather than adapting to variations within a single task.
2. In the setting of online RL, task labels are actually accessible in the simulator. All environments used in the experiments, such as ML45 and multi-game Atari, can provide task labels for each task. Therefore, balancing multi-task losses without task labels is not necessary. Additionally, there are methods that first learn a multi-task label-conditioned policy online and then distill it into a context-based policy [3].
3. The technical contribution is not strong. Compared to AMAGO [2], which proposes the multi-task RL framework using context-based Transformers and already achieves great results on these benchmarks, the contribution of this paper (classification-based actor-critic loss, which exists in the multi-task RL literature) is incremental.
4. According to the literature, the paper lacks comparisons with many baseline methods. The experiments only present ablation studies of the proposed classification loss.
5. Open access to the code is claimed in the checklist. But there is no supplementary or external link provided to access the code. I also cannot find a discussion of limitations, which is claimed in the checklist.

[1] Generalization to New Sequential Decision Making Tasks with In-Context Learning, 2023
[2] AMAGO: Scalable In-Context Reinforcement Learning for Adaptive Agents, 2024
[3] In-Hand Object Rotation via Rapid Motor Adaptation, 2022

**Questions:**

Please see the Weaknesses.

**Limitations:**

I cannot find a discussion of limitations.

---

> ### Author Rebuttal · Authors · 2024-08-07
>
> Thank you for your review. We will discuss your concerns below.
>
>
> > Multi-task RL using in-context adaptation without task labels is not new. Some recent works [1,2] learn in-context meta-RL policies that adapt to diverse tasks
>
> We are focused on end-to-end online RL while Raparthy et al. belongs to a different category of in-context imitation learning methods that is cited on line 90 of our paper.
>
> To expand on this discussion, there are two main differences between (supervised) “in-context” imitation learning (ICIL) and context-based meta-RL.
>
> 1) ICIL typically relies on a dataset of expert demonstrations collected by single-task RL agents. This removes the opportunity for positive transfer between tasks during data collection. Many examples of large-scale generalist policy methods distill single-task experts into a multi-task policy, which highlights the difficulty of online multi-task RL even after many years of study.
>
> 2) ICIL approaches in-context learning much like few-shot prompting in language models, where we select a prompt demonstration and ask the model to continue this behavior over the following inputs. Meta-RL agents begin from a blank sequence and collect examples that will maximize the overall return. IL on aggregated single-task datasets can make it challenging to explore a new environment by creating a mismatch between the task prior of the multi-task policy and the single-task policies it is trying to imitate [1]. In other words, the multi-task policy may be imitating behavior that is not adaptive, but it lets us generalize from demonstrations.
>
> AMAGO is not a multi-task method, which we will discuss below.
>
> [1] *Offline Meta Reinforcement Learning - Identifiability Challenges and Effective Data Collection Strategies*, Dorfman et al., 2021
>
>
> > In the setting of online RL, task labels are actually accessible in the simulator … Therefore, balancing multi-task losses without task labels is not necessary.
>
> This perspective treats current benchmarks as an end goal instead of a step towards a more general method. The ultimate goal of adaptive RL is to automatically generalize to very large numbers of tasks. Methods that rely on task labels scale with the number of tasks (gradient editing, separate task heads, etc.). So while it’s true that we can always find the ground-truth task label in the gym environment of common benchmarks like ML45 and Atari, our work ignores this information in hopes of researching a more general method that can extend past popular benchmarks. We do not want to rely on knowing the identity of a new task at test-time, because this removes chances of generalization to unseen tasks. And we would like to scale to more open-ended domains where the total task set is so diverse it is unclear how to one-hot label / count tasks at train-time. Meta-learning without task labels is capable of these goals in theory while multi-task RL with labels is not.
>
> This argument is covered in the main text, including in lines 52-55, 135-140, and 255-261, but will add additional discussion in Section 2.
>
>
> > AMAGO [2], which proposes the multi-task RL framework using context-based Transformers and already achieves great results on these benchmarks, the contribution of this paper … is incremental.
>
> **Following the terminology in this paper, AMAGO is not a multi-task method**. It learns from many procedurally generated versions of a single task, and uses its Transformer to identify and adapt to those subtle variations. AMAGO evaluates on single-task POPGym, single-task Meta-World (ML1) and so on. It does not attempt ML45 or the multi-task POPGym setting introduced here. The green curves in our figures (“Dep. / Dep.”) represent an AMAGO-style optimization step. We do not claim our method is fundamentally new, but we make an observation that an increasingly common implementation detail in recent work has a simple justification in an online multi-task setting, which unlocks the (very non-incremental) performance improvement in our results.
>
>
> > the paper lacks comparisons with many baseline methods. The experiments only present ablation studies of the proposed classification loss.
>
> The high-level approach of memory-based meta-RL is simple but does not allow for many changes, so most of the differences between methods in this area are implementation-specific. We are evaluating changes in training objectives and ensure a fair ablation of those changes by holding all other details fixed. Our experiments cover a variety of domains, model sizes, and sequence lengths. We provide external reference scores for context where possible (Figures 3, 4, and 5), but some of these settings are rarely (if ever) attempted. Tuning hyperparameters to measure state-of-the-art performance across different conceptual approaches on these benchmarks is not the goal of our work.

---

> > ### Author Response · Authors · 2024-08-13
> >
> > Dear reviewer Ypn5,
> >
> > We have received a reply from the other three reviewers, who maintain their scores and recommend accepting our paper. If you have any additional questions or feel that we can do more to address your initial concerns, please let us know, and we will do our best to get back to you before the end of the author discussion period.
> >
> > Thanks,
> > Authors

---

> > ### Comment · Reviewer_Ypn5 · 2024-08-13
> >
> > Thank you for your response and the clarifications regarding the differences between offline meta-imitation learning and online meta-RL.
> >
> > I appreciate the explanation that AMAGO is not a multi-task method. Both AMAGO and the proposed method are online meta-RL methods. It appears that this work builds on AMAGO, with the novelty primarily in the modification of the actor-critic loss. Given this, my initial concern about baseline selection persists. If the primary contribution of this work is the novel actor-critic loss, it would be valuable to demonstrate its efficacy on more commonly used architectures in online meta-RL, not just within the AMAGO framework. There are numerous online meta-RL methods [1,2,3] that utilize simpler architectures without relying on the specific Transformer-based design shown in Figure 2. On the other hand, if the contribution lies in the combination of the loss function and the Transformer architecture, it becomes crucial to compare with these online meta-RL methods to clearly highlight the advantages of your approach. I believe that the implementation of these baselines is not overly specific, as they have been successfully applied across diverse domains in their original papers.
> >
> > References:
> > [1] VariBAD: A Very Good Method for Bayes-Adaptive Deep RL via Meta-Learning, 2020.
> > [2] Efficient Off-Policy Meta-Reinforcement Learning via Probabilistic Context Variables, 2019.
> > [3] Improving Context-Based Meta-Reinforcement Learning with Self-Supervised Trajectory Contrastive Learning, 2021.

---

> > > ### Author Response · Authors · 2024-08-13
> > >
> > > Thank you for the reply. We hope we can give you a bit more context on this concern:
> > >
> > > - We are only using AMAGO as an example of the simplest RL^2 pure memory strategy that relies entirely on the RL update to drive meta-learning. This framework adds the fewest extra meta-RL components and lets us focus on the bottleneck of the actor/critic objective without introducing other factors. We would not say this technique relies on a specific architecture. If anything the architecture in Figure 2 is simpler than many context-based meta-RL techniques: All we have is a sequence model and two outputs. Transformers are a strong choice for memory if we can train them stably, and AMAGO happens to include some extra details that make training the larger models used here more practical.
> > >
> > > - We are trying to push this technique beyond the familiar toy continuous control benchmarks. Our results immediately jump to Meta-World ML45, which is essentially the upper limit of established meta-RL benchmarks in terms of scale (the methods you have cited here are not evaluating ML45). From there we move to problems that require longer memory (POPGym) and pixel-based learning (Procgen). This is why we cannot provide many comparisons with prior work, although we’ve included or run extra baselines when possible.
> > >
> > > - It’s possible in theory to add the actor/critic change to the codebases of several more specialized meta-learning approaches and run the same direct ablation we have done with RL^2/AMAGO. Aside from the computational cost of repeating the comparison in every domain, there are some technical and practical reasons we have not done this.
> > >
> > >     -  Official implementations of these methods can be quite focused on continuous control and the classic meta-RL locomotion benchmarks.
> > >
> > >     - variBAD and most variants of RL^2 use an on-policy base RL update (usually TRPO or PPO). The classification-style value net can still be used but the details would be different. We would also need a different policy update to make the actor resistant to the scale of returns.
> > >
> > >     - The PEARL [2] (or TCL-PEARL [3]) task exploration strategy is not necessary in these domains because we can adapt quickly within the context of a single episode using memory. When you take this aspect of PEARL away and replace the sequence model with something that is not invariant to temporal order you get off-policy RL^2, which we’ve thoroughly evaluated.
> > >
> > >     - It is very non-trivial to extend the dynamics modeling of methods like variBAD (and others) to pixel-based environments. We don’t doubt that would work (and it would be interesting to find out what extra details are needed to bring accurate pixel modeling to meta-RL / MTRL) but any effort to do this fairly would quickly turn into a Dreamer-like engineering problem and might be a paper on its own.
> > >
> > >     - Adding Transformers to older meta-RL methods designed to train small RNNs is a significant change, even if the final architecture is not much more complex. The AMAGO paper describes this in detail. The model sizes evaluated in our existing results are much larger than those used in these other codebases.
> > >
> > > **In summary:** We’ve chosen to evaluate our change in the vanilla sequence model framework where it is most directly impactful. The scale and novelty of our experimental domains limits how many reference comparisons we can include in the results, so we focus on creating a fair self-comparison. It’s possible to repeat the same comparison on top of other codebases, but in practice this involves much more than replacing one loss function with another.

---

### Author Rebuttal · Authors · 2024-08-07

We would like to thank all the reviewers for their comments, and we will respond to individual questions and concerns below.


Several reviewers asked for an expanded discussion of our method’s limitations, which we will add to our conclusion. The main technical limitation of our technique is that it does not address all of the challenges of optimizing distinct task objectives. These challenges are not unique to RL and are the focus of many prior works, as discussed in Section 2. Instead, we are minimizing an additional challenge that value-based RL algorithms can introduce, which is that we are unintentionally rescaling our learning objective(s) throughout training according to a schedule that is difficult to predict or control. We demonstrate a simple approach that lets us manage this issue without introducing explicit task knowledge that would make our agents less applicable to general settings.

Another limitation is that our experiments evaluate the ability to generalize across unseen variants of many different tasks but do not evaluate the ability to generalize to entirely new tasks.  This kind of non-parametric task variation requires large training sets of unique tasks that are beyond the reach of many current benchmarks. We think the path towards this goal requires the ability to learn from the small disjoint task sets currently available. Existing methods struggle to do this without multi-task optimization techniques that rely on our ability to label tasks. We aim to provide a simple way to scale a standard meta-learning approach to increasingly diverse training sets and enable new research on complex domains beyond typical toy problems.

---

### Comment · Area_Chair_QM66 · 2024-08-11
**Code submission**

The authors have submitted their experimental code to me. This year’s guidelines for the review/discussion period specify that external code links can only be submitted to ACs.


The code is well-organized and includes instructions for reproducing the experiments. It is primarily based on the AMAGO codebase ("AMAGO: Scalable In-Context Reinforcement Learning for Adaptive Agents" by Grigsby et al. (2024)).


I hope this clarifies the reviewers' question regarding code access.

Best regards,

AC

---

### Decision · Program_Chairs · 2024-09-25

**Decision:**

Accept (poster)

**Comment:**

This paper tackles the challenge of scaling meta-reinforcement learning (meta-RL) to manage multiple tasks without explicit task labels. Building upon recent meta-RL techniques, it presents a method that transforms both the actor and critic objectives into classification problems, separating optimization from the scale of returns. The method's effectiveness is validated on various benchmarks, including Meta-World ML45, Multi-Game Procgen, and Multi-Task POPGym, demonstrating notable improvements.

All reviewers indicated high confidence in their reviews, with 3 out of 4 reviewers recommended to accept the paper, and reviewer Ypn5 leaning rejection. However, I find that most of Ypn5's concerns are adequately addressed during the rebuttal. In particular, I agree with the authors that further extending experiments with more baseline comparisons requires a large amount of compute and non-trivial modifications to existing methods that are likely beyond the scope of a conference paper. The method as it stands represents a good contribution that future works may build upon.

To authors: from my reading of the paper, the presentation could be further improved. The paper assumes a high-level of familiarity with previous works and is dense at times. For instance, there are multiple under-defined terms such as "two-hot", and $d_{i-1}$ in $ (o_i, a_{i−1}, r_{i−1}, d_{i−1})$ is only defined in the figure but not the main text. Additionally, please carefully revise the final version to include the discussion with the reviewers.